# MUSTAFAR: Promoting Unstructured Sparsity for KV Cache Pruning in LLM Inference

**Donghyeon Joo[1], Helya Hosseini[1], Ramyad Hadidi[2], Bahar Asgari[1]**
[1]Department of Computer Science, University of Maryland, [2]d-Matrix
{dhjoo98,helia,bahar}@umd.edu, rhadidi@d-matrix.ai

## Abstract

We demonstrate that unstructured sparsity significantly improves KV cache compression for LLMs, enabling sparsity levels up to 70% without compromising accuracy or requiring fine-tuning. We conduct a systematic exploration of pruning strategies and find per-token magnitude-based pruning as highly effective for both Key and Value caches under unstructured sparsity, surpassing prior structured pruning schemes. The Key cache benefits from prominent outlier elements, while the Value cache surprisingly benefits from a simple magnitude-based pruning despite its uniform distribution. KV cache size is the major bottleneck in decode performance due to high memory overhead for large context lengths. To address this, we use a bitmap-based sparse format and a custom attention kernel capable of compressing and directly computing over compressed caches pruned to arbitrary sparsity patterns, significantly accelerating memory-bound operations in decode computations and thereby compensating for the overhead of runtime pruning and compression. Our custom attention kernel coupled with the bitmap-based format delivers substantial compression of KV cache up to 45% of dense inference and thereby enables longer context lengths and increased tokens/sec throughput of up to $2.23\times$ compared to dense inference. Our pruning mechanism and sparse attention kernel is available at `https://github.com/dhjoo98/mustafar`.

## 1 Introduction

In the age of Large Language Models (LLMs), advances in the machine learning domain [41, 2, 6] and the fast and efficient computing systems [21, 35] have led to the emergence of highly capable LLMs that can summarize a book [22], write a compelling story [18], code a library [53], and generally reason over longer contexts than ever before [7]. As LLMs are increasingly tasked with processing longer sequences, the memory overhead associated with key-value (KV) caching has emerged as a critical bottleneck to scaling context length.

Prior work has approached the challenge of KV cache memory overhead through techniques such as quantization [30, 15, 48, 52], low-rank approximation [47, 4, 37, 50, 26], token-wise eviction [51, 29, 25, 8, 1, 11], and structured pruning (e.g., channel-wise removal [44, 31]). The need to improve individual compression techniques has become increasingly important, especially as joint applications of multiple methods, such as pruning combined with token eviction [44], quantization with token-wise eviction [52], and low-rank approximation with quantization [4], gain popularity. However, previous work on KV cache pruning have been limited to structured pruning, primarily due to the difficulty of efficiently leveraging finer-grained (i.e., unstructured) sparsity during execution. Effective pruning of the KV cache entails two core challenges: (1) achieving substantial reduction in KV cache size while preserving model accuracy, and (2) ensuring that the runtime pruning and compression processes are sufficiently efficient (i.e., the associated overhead must not outweigh the latency gains introduced by the resulting sparsity).

39th Conference on Neural Information Processing Systems (NeurIPS 2025).

In this paper, we find that removing any constraint on the sparsity pattern, effectively unstructured sparsity can ensure that compressed KV cache perform with minimal model accuracy degradation while being pruned to a higher sparsity. In Section 2 (green region of Figure 1), we first present our journey to find the optimal pruning algorithm for the key and value cache, based on the element magnitude distributions of the KV cache. We explore the feasibility of various pruning algorithms on both KV cache to conclude that applying a simple per-token magnitude-based pruning on both Key and Value caches is capable of preserving the model accuracy at a high sparsity, while also demonstrating strong compatibility with orthogonal compression techniques.

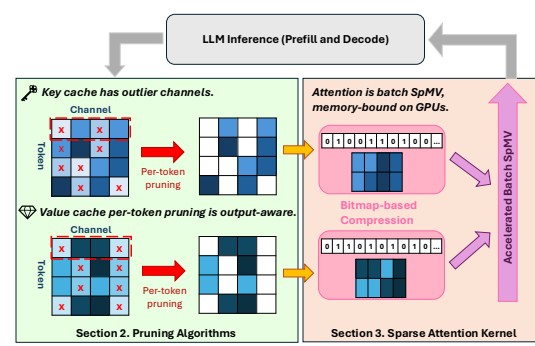

Figure 1: High-level overview of Mustafar. Green region describes the pruning algorithm of Section 2, pink region describes the custom sparse attention kernel of Section 3.

Section 3 (pink region of Figure 1) discusses the next step: having induced sparsity in the KV cache, the challenge becomes leveraging the unstructured sparsity to reduce memory footprint and accelerate computation. To this end, we adopt a bitmap-based sparse format that serves two purposes. First, the bitmap enables maximal compression of matrices with arbitrary sparsity patterns. Second, this maximal compression of matrix operands translates into computational speedup of the attention operation, which is severely memory-bound on GPUs. Alongside the sparse format, we introduce the custom attention kernel tailored to operate on the bitmap-based sparse format. We see that the speedup of our attention kernel overshadows the latency introduced by runtime pruning and compression, meanwhile effectively compressing the KV cache to high sparsity with minimal accuracy degradation.

In summary, we demonstrate that adopting unstructured sparsity in the KV cache without imposing constraints on the pruning pattern enables higher degrees of sparsity while preserving model accuracy. Furthermore, we introduce the necessary computational tools to support unstructured sparsity efficiently, ensuring that the derived high sparsity leads to gains in memory compression and end-to-end inference throughput.

## 2   Pruning Algorithm for Unstructured Sparsity

> **Question**: Does removing structural constraints in KV cache pruning allow for higher sparsity while preserving model accuracy more effectively than structured pruning methods?

We explore the potential unstructured sparsity on KV cache pruning by considering the two factors for Key and Value cache pruning: pruning direction and output-awareness. **Pruning Direction** refers to the axis along which sparsity is induced when selecting elements for removal. Since both the Key and Value caches are represented as matrices with dimensions $[tokens \times channels]$, we consider two primary pruning directions: per-channel pruning, which determines target sparsity across each channel (i.e., across tokens for each channel), and per-token pruning, which determines target sparsity across each token's cache (i.e., across model dimensions for each token). **Output-Awareness** refers to the use of a scoring metric that serves as a proxy for estimating each element's contribution to the operation's output. Commonly employed in LLM weight pruning [38] and structured KV cache pruning [44], this technique involves computing a score for each pruning unit such as a channel or an element by taking the product of the corresponding element with its associated input. This approach effectively captures the element's influence on the final output, guiding more informed pruning decisions. For a fair and effective comparison between pruning strategies, we uniformly employ a **local dense window**, where the recent 32 tokens remain untouched during the decode phase. Previous works [51, 44] have shown that this is effective in preserving model accuracy, meanwhile being small enough in size to not severely impact the compression.

## 2.1 Pruning Key Cache

In deciding the pruning direction, we build on top of the observation of KIVI [30], that Key cache exhibits distinct channel-wise outliers, where "channel" refers to the head dimension (Figure 2a). This leads us to focus on per-token pruning for key cache, as it can effectively capture the elements in the outlier channel.

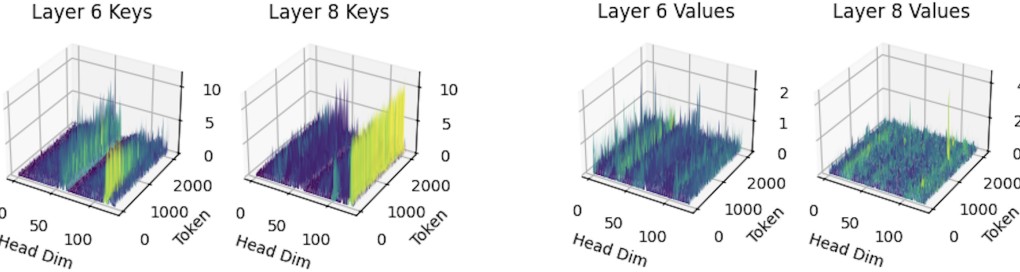

(a) Magnitude distribution of Key cache         (b) Magnitude distribution of Value cache

Figure 2: Visualization of the KV cache in LLaMA-2 7B. Color intensity indicates element magnitude. The figure was generated using the visualization code from KIVI [30].

Based on the same observation to perform structured pruning of individual channels, ThinK [44] incorporates output-awareness by using a per-channel score of the accumulation of last 32 query, multiplied by each channel. To this end we compare the accuracy of ThinK [44], per-token magnitude-based unstructured pruning, and output-aware unstructured pruning of our design.

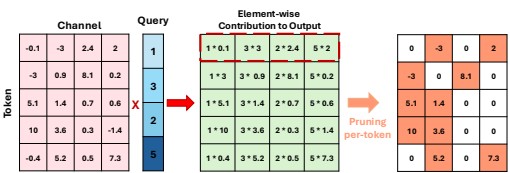

Figure 3: Per-token, output-aware pruning of Key cache

Figure 3 elaborates the per-token output-aware unstructured pruning score of Key cache. The element-wise $L_1$ accumulation of the current and next 31 Query vector (blue) is multiplied element-wise across each token's key vector (pink) to derive the pruning score (green). The absolute value of the score element in the corresponding position of each Key cache element is used to decide the elements to be pruned within a token's Key vector. In other words, we formulate the per-token output-aware unstructured pruning score $S$ of a Key cache $K$ to be:

$$S = |K| \odot broadcast\left(\sum_{t=T}^{T+31} |Q_t|\right), \quad \text{where } Q_t \text{ is the query at time } t$$

For Group Query Attention (GQA) [2], where multiple queries correspond to a KV cache pair, we sum the pruning score of all queries mapped to each KV cache.

Table 1: Comparison of ThinK [44] structured pruning, per-token magnitude-based unstructured pruning, and per-token output-aware unstructured pruning on LongBench [3] with Llama-3-8B-Instruct Key cache. $K_s$ denotes Key cache sparsity.

| Task | Dense | $K_s = 0.5$ | | | $K_s = 0.7$ | | |
|---|---|---|---|---|---|---|---|
| | | ThinK (Structured) | Unstructured Output-aware | Unstructured Magnitude | ThinK (Structured) | Unstructured Output-aware | Unstructured Magnitude |
| Average | 43.19 | 38.53 | **43.23** | 42.84 | 26.55 | **42.13** | 41.55 |
| SingleDoc QA | 36.66 | 35.61 | 36.57 | **36.90** | 25.26 | **35.78** | 35.53 |
| MultiDoc QA | 36.09 | 34.99 | **35.92** | 35.77 | 29.75 | **35.55** | 35.40 |
| Summarization | 26.75 | 24.96 | **26.87** | 26.45 | 17.70 | 25.16 | **25.18** |
| Few-shot | 68.96 | 66.54 | **68.82** | 68.75 | 44.88 | 67.22 | **67.84** |
| Synthetic | 37.25 | 35.50 | **37.00** | 36.75 | 16.86 | **35.25** | 35.00 |
| Code | 55.58 | 29.56 | **56.61** | 54.14 | 19.15 | **56.19** | 51.47 |

In Table 1, we compare Llama-3-8B-Instruct accuracy of different pruning methods on LongBench [3]. For structured pruning, we see that even at a moderate sparsity, model accuracy retention is dismal compared to pruning to an unstructured sparsity pattern. Notably, unstructured pruning is capable of outperforming structured pruning even without the memory footprint of pruning scores involved with output-awareness. Applying output-awareness to unstructured pruning results in a slight improvement in the LongBench total average score, while individual task performance is mixed with each method outperforming the other on different tasks.

> **Key Cache Verdict:** While the existence of outlier channels with exceptionally high magnitudes show promise for per-channel structured pruning, unstructured sparsity achieves higher accuracy at greater sparsity levels, even without output-awareness.

## 2.2 Pruning Value Cache

As shown in Figure 2b, Value cache exhibits more uniform distribution of activations, making it challenging to apply the same channel-wise pruning without incurring substantial degradation in model accuracy. This difficulty has led recent Value cache pruning approaches to be more susceptible to accuracy degradation.

With no discernible outliers in certain direction, we explore all possible combinations of (pruning direction, magnitude/output-aware) pairs. However, we are able to rule out per-token output-aware pruning, as the attention formulation $AttentionScore \times Value$ involves a multiply-and-accumulate operation along the token dimension. As seen in Figure 4, every element of a token's Value cache is multiplied by the same element of the attention score, with each element's impact on the output proportionate to the magnitude of each value. **That is, for Value cache pruning, per-token magnitude-based pruning is already output-aware**. For per-channel pruning, we prune each channel to the target sparsity in groups of 32 tokens, for compatibility with the local window size. For

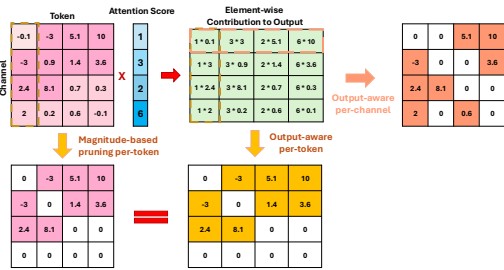

Figure 4: Output-aware per-channel (red) and magnitude-based per-token (pink) pruning of Value cache. Magnitude-based per-token pruning is equal to output-aware per-token pruning (yellow).

per-channel output-aware pruning, we accumulate the current and subsequent 31 attention score $\alpha$ of each token, which is then element-wise multiplied to the corresponding Value Cache $(V)$ element. The following formula describes the pruning score $S$ of per-channel output-aware pruning:

$$S = |V| \odot broadcast \left( \sum_{t=T}^{T+31} |\alpha_t| \right), \quad \text{where } \alpha_t \text{ is the attention score at time } t$$

Table 2: Comparison of ThinK [44] structured pruning, per-channel magnitude-based unstructured pruning, per-channel output-aware unstructured pruning, and per-token magnitude-based pruning on LongBench [3] with Llama-3-8B-Instruct Value Cache. $V_s$ denotes Value cache sparsity.

| Task | Dense | $V_s = 0.5$ | | | | $V_s = 0.7$ | | | |
|---|---|---|---|---|---|---|---|---|---|
| | | ThinK (Structured) | Magnitude (Per-channel) | Output-aware (Per-channel) | Magnitude (Per-token) | ThinK (Structured) | Magnitude (Per-channel) | Output-aware (Per-channel) | Magnitude (Per-token) |
| Average | 43.19 | 38.45 | 42.50 | 42.84 | **43.04** | 30.60 | 41.69 | 42.67 | **42.78** |
| SingleDoc QA | 36.66 | 34.92 | 36.56 | 36.24 | **36.75** | 25.05 | 36.11 | 36.05 | **36.96** |
| MultiDoc QA | 36.09 | 34.74 | 35.45 | 36.07 | **36.22** | 23.90 | 35.11 | **36.20** | 35.82 |
| Summarization | 26.75 | 23.31 | 24.74 | 25.79 | **26.34** | 20.41 | 22.72 | 24.75 | **25.19** |
| Few-shot | 68.96 | 67.18 | 67.66 | 68.65 | **68.91** | 60.16 | 67.39 | **68.23** | 68.08 |
| Synthetic | 37.25 | 35.43 | **38.31** | 37.00 | 36.25 | 29.63 | **38.75** | 37.25 | 35.50 |
| Code | 55.58 | 31.97 | 55.07 | 55.57 | **55.77** | 20.85 | 52.65 | 56.17 | **57.62** |

As shown in the Table 2, we first see that applying structured pattern to Value cache pruning incurs significant accuracy degradation even in 50% sparsity. This is concurrent with ThinK [44] findings, which points to 30% sparsity as the upper-bound on acceptable accuracy. In contrast, per-token

magnitude pruning is capable of preserving model accuracy even at 70% sparsity. For per-channel pruning, we see that incorporating output-awareness boasts model accuracy retention almost to the level of per-token pruning. However, we prefer per-token magnitude-based pruning for the following two reasons. First, output-aware per-channel value cache pruning requires access to the attention score which requires additional recomputation when used alongside FlashAttention [6], where the full attention score matrix does not materialize in the global memory. Second, per-token magnitude-based pruning allows smooth compatibility with orthogonal compression method token-wise eviction [24, 51], where the retained token's KV cache can be pruned individually. We examine the accuracy of joint application in Section 4.2.

> **Value Cache Verdict**: All unstructured pruning methods explored outperform structured pruning. Among unstructured pruning methods, token-wise pruning, which is inherently output-aware by matrix multiplication formulation, best preserves model accuracy even at high sparsity levels. While channel-wise pruning with output-awareness can achieve comparable accuracy, token-wise pruning offers advantages in both efficiency and modularity.

With the two verdicts in Key and Value caches, on Table 3 we finally validate the model accuracy retention of per-token magnitude-based pruning with both Key and Value caches pruned. Not only can Value cache be pruned to high sparsity with unstructured sparsity, but both KV cache can be pruned to 70% sparsity while showing similar or better accuracy than Key-only 50% structured pruning of ThinK [44]. In Appendix A.1, methodology of this section is applied on Llama-2 7B to reinforce the effectiveness of per-token magnitude-based KV cache pruning.

Table 3: Longbench Score of Llama-3-8B-Instruct and Mistral-7B-Instruct-v0.2 with KV Cache Per-Token Magnitude-based Pruning.

| Task | Llama-3-8B-Instruct | | | Mistral-7B-Instruct-v0.2 | | |
|---|---|---|---|---|---|---|
| | **Dense** | $K_s = 0.5$ $V_s = 0.5$ | $K_s = 0.7$ $V_s = 0.7$ | **Dense** | $K_s = 0.5$ $V_s = 0.5$ | $K_s = 0.7$ $V_s = 0.7$ |
| Average | 43.19 | 42.65 | 40.96 | 42.65 | 42.30 | 40.95 |
| SingleDoc QA | 36.66 | 36.67 | 35.28 | 36.21 | 36.22 | 36.08 |
| MultiDoc QA | 36.09 | 36.23 | 35.11 | 29.93 | 30.42 | 29.40 |
| Summarization | 26.75 | 26.05 | 23.57 | 28.10 | 27.77 | 26.72 |
| Few-shot | 68.96 | 68.18 | 66.10 | 66.68 | 66.70 | 66.24 |
| Synthetic | 37.25 | 36.00 | 34.13 | 44.85 | 41.92 | 36.13 |
| Code | 55.58 | 54.50 | 53.49 | 54.98 | 54.83 | 53.84 |

## 3  Sparse Attention Kernel

Our findings establish that unstructured sparsity offers superior sparsity ratios over structured sparsity while preserving accuracy. In turn, a crucial contribution of Mustafar is to leverage this advantage to enable high compression efficiency while minimizing the latency overhead of runtime pruning and compression. Prior compression methods such as quantization, structured pruning, and token eviction reduce matrix dimensions or element bitwidths. In terms of efficiency, speedup from the reduced size of dense matrix operands compensates for the additional latency introduced by compression (i.e. pruning score computation, quantization). In contrast, unstructured sparsity with no regular reduction in dimensions or element bitwidth demands a different approach.

Mustafar is motivated by the observation that attention operations in the autoregressive decode stage, the $Query \times Key^T$ and $Attention\ Score \times Value$ computations are batch (different heads) of matrix-vector products (MVs) that are significantly memory-bound on GPUs compared to the prefill stage. To exploit this property, we extend the bitmap-based sparse format of Coruscant [20] as shown in Figure 5a to maximally compress the pruned KV cache. It consists of compressed tiles corresponding to a $1 \times 64$ column of the pruned cache. Per-tile bitmap of 64 bits is used to represent the position of non-zeros, and tile offset is used to address the correct position of each tile's starting non-zero. Pruning and compression are performed on-the-fly, with compression accelerated on GPU with a Triton kernel, and attention is computed directly on the compressed representation with a custom CUDA kernel that performs batch SpMV on the bitmap-based sparse format. Memory-bound decode-phase attention is accelerated by reducing the data movement from global memory to GPU Streaming Multiprocessors.

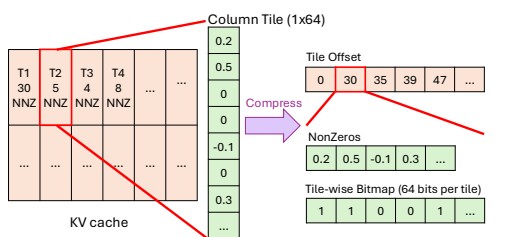
(a) Coruscant [20] bitmap-based sparse format

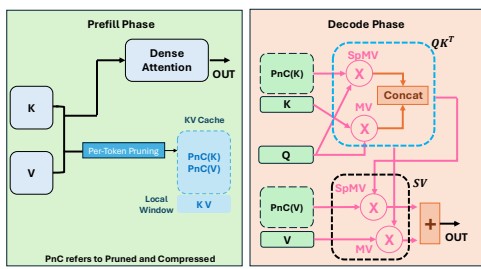
(b) Mustafar attention kernel formulation

Figure 5: Overview of Mustafar sparse attention kernel. In (b), multi-head, softmax, and normalization are omitted for simplicity.

Figure 5b and Algorithm 1 presents the Mustafar sparse attention kernel. KV cache generated in prefill stage is pruned and compressed before the start of decode stage, therefore compatible with prefill FlashAttention [6]. KV cache generated in decode stage is kept as-is (dense) while it is within the local window, then pruned and compressed afterwards. This entails the attention computations in the decode stage to be reformulated into two parts: SpMV for compressed KV cache (line 2 and 5 of Algorithm 1) and dense MV for the KV cache within the local window (line 1 and 5 of Algorithm 1).

---

**Algorithm 1** Decode Phase Attention with Dense Local and Compressed KV Caches

---

**Input:** Query $\mathbf{Q}_t \in \mathbb{R}^d$; Local KV cache $\mathbf{K}_L, \mathbf{V}_L \in \mathbb{R}^{d \times N_d}$, where $N_d$ is size of local window in tokens; Compressed KV cache $\mathbf{K}_C, \mathbf{V}_C \in \mathbb{R}^{d \times N_s}$, where $N_s$ is number of compressed tokens.

**Attention Score Computation**
1: $\mathbf{S}_L \in \mathbb{R}^{1 \times N_d} \leftarrow \mathbf{Q}_t \mathbf{K}_L$            *Dense local window attention score*
2: $\mathbf{S}_C \in \mathbb{R}^{1 \times N_s} \leftarrow \mathbf{Q}_t \mathbf{K}_C$         *Sparse attention score over compressed KV cache*
3: $\mathbf{S}_t \in \mathbb{R}^{1 \times (N_s + N_d)} \leftarrow \text{softmax}\big(\text{concat}(\mathbf{S}_C, \mathbf{S}_L)\big)$       *Full attention score*

**Output Computation**
4: $[\mathbf{S}_C, \mathbf{S}_L] \leftarrow \text{split}(\mathbf{S}_t; N_s, N_d)$          *Partition attention score*
5: $\mathbf{O}_t \in \mathbb{R}^d \leftarrow \mathbf{V}_C \mathbf{S}_C^\top + \mathbf{V}_L \mathbf{S}_L^\top$         *Final output vector*
**Return $\mathbf{O}_t$**

---

Mustafar SpMV kernel follows the load-as-compressed, compute-as-dense paradigm adopted by FlashLLM [43], SpInfer [9], and Coruscant [20], which target sparse matrix–dense matrix multiplication in LLM weight projection layers. The compressed KV cache is loaded from GPU global memory into registers in its compressed form, decompressed into shared memory, and then used for tile-wise dense computation. We evaluate the performance of the Mustafar attention kernel and quantify the runtime overhead of pruning and compression in Section 4.3. We further detail the formulation of the SpMV kernel, as well as the management of the compressed KV cache in Appendix C.

## 4 Evaluation

**Methodology**: We evaluate Mustafar on two aspects: Accuracy and Efficiency. For accuracy evaluation, we use tasks from LongBench [3] to test the accuracy retention of per-token magnitude-based pruning of KV cache. We evaluate on three models: Llama-2-7B [40], Llama-3-8B-Instruct [12], and Mistral-7B-Instruct-v0.2 [19]. We also explore the impact of Mustafar when jointly used with orthogonal compression techniques, KV cache quantization KIVI [30] and token-wise eviction H2O [51]. For efficiency evaluation, we evaluate the impact on KV cache compression and computational latency with Llama-2-7B and Llama-3-8B-Instruct. Efficiency evaluation is tested on NVIDIA RTX 6000ADA GPU and measured with NVIDIA Nsight Profiling Tool. Additionally, we provide accuracy evaluation on RULER [17] benchmark in Appendix A.3, accuracy comparison of Mustafar's unstructured sparsity with 2:4 semi-structured sparsity in Appendix B, and additional kernel throughput evaluation in Appendix C.3.

## 4.1 LongBench Results

Table 4 shows the extended LongBench evaluation of Mustafar per-token magnitude-based pruning with comparison to dense model and ThinK [44]. Under the same Key cache sparsity, unstructured nature of Mustafar constantly achieves higher accuracy than structured sparsity on ThinK [44] across all tasks. A key advantage of unstructured pruning is its ability to effectively prune the Value cache with minimal accuracy degradation, which structured pruning has struggled to achieve. Even under high sparsity 70% for both the Key and Value caches, unstructured pruning (yellow) consistently outperforms ThinK's Key-only 50% structured pruning (pink) on LLaMA-3 8B and Mistral 7B, and achieves comparable accuracy on LLaMA-2-7B.

Table 4: Mustafar accuracy with Llama and Mistral on LongBench

| KV Sparsity | Single-Document QA | | | Multi-Document QA | | | Summarization | | | Few-shot Learning | | | Synthetic | | Code | | Avg. |
|---|---|---|---|---|---|---|---|---|---|---|---|---|---|---|---|---|---|
| | NrtvQA | Qasper | MF-en | HotpotQA | 2WikiMQA | Musique | GovReport | QMSum | MultiNews | TREC | TriviaQA | SAMSum | PCount | PRe | Lcc | RBP | |
| **Llama-3 8B Instruct** | | | | | | | | | | | | | | | | | |
| Dense | 23.39 | 43.38 | 43.22 | 46.39 | 38.66 | 23.22 | 29.91 | 22.56 | 27.77 | 74.50 | 90.28 | 42.11 | 4.50 | 70.00 | 57.11 | 54.05 | **43.19** |
| ThinK0.5 | 22.38 | 40.96 | 43.48 | 44.01 | 38.37 | 22.59 | 26.61 | 22.20 | 26.08 | 74.00 | 88.83 | 36.79 | 6.00 | 65.00 | 27.95 | 31.17 | **38.53** |
| K0.5 V0.0 | 23.40 | 43.68 | 43.63 | 46.00 | 38.60 | 22.72 | 29.39 | 22.33 | 27.64 | 74.50 | 90.66 | 41.09 | 5.00 | 68.50 | 55.89 | 52.39 | **42.84** |
| ThinK0.7 | 17.58 | 27.40 | 30.80 | 40.59 | 29.50 | 19.16 | 18.13 | 17.28 | 17.70 | 34.00 | 83.09 | 17.56 | 4.71 | 29.00 | 17.88 | 20.42 | **26.55** |
| K0.7 V0.0 | 22.91 | 42.36 | 41.33 | 45.53 | 38.50 | 22.16 | 26.63 | 21.90 | 27.00 | 73.00 | 90.83 | 39.68 | 4.50 | 65.50 | 51.94 | 50.99 | **41.55** |
| K0.0 V0.5 | 23.80 | 43.14 | 43.32 | 46.28 | 39.42 | 22.97 | 29.18 | 22.70 | 27.13 | 74.50 | 90.50 | 41.74 | 5.00 | 67.50 | 57.23 | 54.30 | **43.04** |
| K0.0 V0.7 | 24.19 | 42.78 | 43.92 | 45.82 | 39.11 | 22.53 | 26.92 | 22.52 | 26.12 | 74.00 | 90.36 | 39.88 | 5.50 | 65.50 | 59.18 | 56.05 | **42.77** |
| K0.5 V0.5 | 23.40 | 46.63 | 42.98 | 46.28 | 39.27 | 23.13 | 28.29 | 22.78 | 27.07 | 74.00 | 90.58 | 39.97 | 5.00 | 67.00 | 55.54 | 53.46 | **42.65** |
| K0.7 V0.7 | 24.10 | 40.85 | 40.88 | 44.93 | 38.03 | 22.36 | 24.02 | 21.90 | 24.78 | 70.50 | 90.04 | 37.77 | 5.25 | 63.00 | 54.12 | 52.86 | **40.96** |
| **Mistral-7B-Instruct-v0.2** | | | | | | | | | | | | | | | | | |
| Dense | 26.76 | 32.51 | 49.36 | 43.49 | 27.48 | 18.81 | 32.95 | 24.36 | 27.00 | 71.00 | 86.23 | 42.80 | 2.89 | 86.81 | 55.89 | 54.07 | **42.65** |
| ThinK0.5 | 24.03 | 26.79 | 46.42 | 38.70 | 24.93 | 15.73 | 32.72 | 24.65 | 27.14 | 71.00 | 85.80 | 41.68 | 2.20 | 73.67 | 48.83 | 47.09 | **39.46** |
| K0.5 V0.0 | 26.38 | 33.08 | 49.20 | 43.90 | 28.57 | 18.65 | 32.47 | 24.21 | 27.05 | 71.00 | 86.28 | 42.66 | 3.00 | 84.23 | 55.72 | 54.16 | **42.56** |
| ThinK0.7 | 19.25 | 21.33 | 36.48 | 27.96 | 20.34 | 14.08 | 29.32 | 22.23 | 25.64 | 70.50 | 78.99 | 29.66 | 2.92 | 54.42 | 34.28 | 31.68 | **32.44** |
| K0.7 V0.0 | 27.02 | 34.37 | 49.26 | 43.77 | 26.37 | 17.45 | 32.05 | 24.09 | 27.43 | 71.00 | 87.19 | 42.30 | 4.65 | 77.24 | 54.26 | 53.06 | **41.97** |
| K0.0 V0.5 | 26.29 | 32.54 | 49.01 | 43.99 | 28.02 | 19.28 | 32.07 | 23.74 | 26.98 | 71.00 | 86.56 | 42.79 | 2.71 | 81.77 | 55.14 | 54.16 | **42.25** |
| K0.0 V0.7 | 26.83 | 31.66 | 49.24 | 44.15 | 27.40 | 18.36 | 30.58 | 23.80 | 26.63 | 71.00 | 86.82 | 42.02 | 3.77 | 76.32 | 55.58 | 54.16 | **41.77** |
| K0.5 V0.5 | 26.90 | 32.99 | 48.76 | 43.90 | 28.90 | 18.45 | 32.24 | 24.09 | 26.99 | 71.00 | 86.68 | 42.41 | 3.20 | 80.64 | 55.51 | 54.15 | **42.30** |
| K0.7 V0.7 | 27.11 | 32.23 | 48.90 | 43.63 | 27.12 | 17.43 | 29.38 | 23.99 | 26.79 | 71.00 | 86.59 | 41.14 | 4.69 | 67.57 | 54.86 | 52.82 | **40.95** |
| **Llama-2 7B** | | | | | | | | | | | | | | | | | |
| Dense | 15.04 | 9.66 | 21.88 | 7.69 | 9.95 | 3.66 | 17.26 | 21.29 | 3.5 | 66.00 | 87.72 | 41.66 | 1.70 | 6.64 | 66.66 | 59.82 | **27.51** |
| ThinK0.5 | 15.57 | 9.96 | 23.31 | 6.50 | 9.62 | 2.77 | 1.84 | 20.16 | 0.38 | 66.00 | 85.53 | 41.48 | 2.04 | 2.79 | 64.77 | 58.36 | **25.69** |
| K0.5 V0.0 | 14.79 | 9.65 | 21.67 | 7.48 | 10.10 | 4.11 | 17.24 | 20.84 | 3.64 | 66.00 | 87.72 | 41.26 | 1.38 | 6.42 | 67.15 | 59.89 | **27.46** |
| ThinK0.7 | 13.76 | 8.16 | 20.59 | 4.53 | 6.24 | 2.23 | 12.96 | 14.88 | 0.01 | 66.00 | 80.48 | 26.95 | 1.77 | 6.93 | 40.73 | 38.97 | **21.57** |
| K0.7 V0.0 | 14.57 | 8.18 | 20.55 | 6.64 | 9.95 | 3.28 | 13.80 | 20.25 | 0.88 | 66.00 | 86.64 | 38.32 | 2.12 | 4.04 | 64.86 | 58.59 | **26.17** |
| K0.0 V0.5 | 15.71 | 10.02 | 21.12 | 7.38 | 9.64 | 3.75 | 16.86 | 21.37 | 2.38 | 66.00 | 87.72 | 41.04 | 1.65 | 6.75 | 66.79 | 60.09 | **27.40** |
| K0.0 V0.7 | 15.57 | 8.98 | 20.97 | 7.33 | 10.14 | 3.82 | 15.40 | 20.77 | 1.83 | 66.00 | 87.72 | 40.69 | 1.40 | 6.50 | 66.12 | 59.57 | **27.05** |
| K0.5 V0.5 | 15.49 | 9.17 | 20.97 | 7.51 | 10.04 | 3.78 | 16.46 | 21.02 | 3.36 | 66.00 | 87.72 | 40.81 | 1.22 | 5.88 | 66.78 | 59.53 | **27.23** |
| K0.7 V0.7 | 13.76 | 7.83 | 19.27 | 6.57 | 10.26 | 3.51 | 8.70 | 20.04 | 0.47 | 64.50 | 86.89 | 36.37 | 1.64 | 3.62 | 63.95 | 56.75 | **25.26** |

## 4.2 Joint Application with Orthogonal KV Cache Compression Techniques

Mustafar's per-token pruning enables seamless integration with orthogonal KV cache compression techniques. We evaluate its effectiveness when combined with token eviction from H2O [51] and KV cache quantization from KIVI [30], using a representative subset of LongBench tasks from each category. H2O application is conducted with Llama-2 7B and KIVI application is conducted with LLaMA-3-8B-Instruct.

### 4.2.1 Joint Application with Token Eviction

H2O [51] retains a fixed budget of recent tokens and critical heavy-hitter tokens. Applying Mustafar to H2O, we retain the same scheme of pruning the KV cache of tokens that exit the local dense window. We configure 10% of KV cache budget each to recent tokens and heavy-hitter tokens. Jointly applied, all heavy-hitter tokens and a part of recent tokens is kept as pruned and compressed. In Table 5, we validate the efficacy of Mustafar's accuracy retention when jointly applied with token eviction, as we see that 50% sparsity in both KV cache retains the dense accuracy with some degradation when pruned to 70% sparsity.

Table 5: LongBench evaluation of Mustafar-H2O joint application on Llama-2-7B

|  | Single-Doc QA
NtrvQA | Multi-Doc QA
HotpotQA | Summarization
GovReport | Few-shot Learning
TREC | Synthetic
Pcount | Code
Lcc |
|---|---|---|---|---|---|---|
| **Full KV cache** | 15.04 | 7.69 | 17.26 | 66.00 | 1.7 | 66.66 |
| **H2O 20% KV Budget** | | | | | | |
| Dense | 12.26 | 8.35 | 7.76 | 64.00 | 1.43 | 64.20 |
| K0.5 V0.0 | 11.83 | 8.47 | 7.65 | 64.00 | 2.08 | 64.72 |
| K0.7 V0.0 | 11.39 | 8.46 | 6.34 | 64.00 | 1.69 | 63.92 |
| K0.0 V0.5 | 12.17 | 8.39 | 6.84 | 64.00 | 1.38 | 64.83 |
| K0.0 V0.7 | 12.39 | 7.79 | 5.81 | 64.00 | 0.76 | 64.88 |
| K0.5 V0.5 | 12.07 | 8.16 | 7.61 | 64.00 | 2.05 | 65.15 |
| K0.7 V0.7 | 12.20 | 8.18 | 5.22 | 64.00 | 1.65 | 63.73 |

### 4.2.2 Joint Application with Quantization

KIVI [30] applies a per-channel quantization of Key cache and per-token quantization of Value cache. Following findings of Harma et al. [14], we first prune each token's KV cache before quantization is performed. However, we note that current Mustafar sparse attention kernel implementation does not support low-bit precision. Therefore, the accuracy measurement was performed on a sparse quantized KV cache. Table 6 shows the performance of Mustafar and KIVI applied together. Similar to joint application with H2O, we see that model accuracy is retained across the tasks for 50% on Key cache, Value cache, as well as both Key and Value caches. We observe a decrease in accuracy at 70% pruning, with Summarization task seeing the most significant drop. However, other tasks, such as Single-Document QA maintain the same performance as naive 16-bit model, suggesting the potential for applying varying degrees of compression tailored to specific tasks.

Table 6: LongBench evaluation of Mustafar-KIVI joint application on Llama-3-8B-Instruct

|  | Single-Doc QA
NtrvQA | Multi-Doc QA
HotpotQA | Summarization
GovReport | Few-shot Learning
TREC | Synthetic
Pcount | Code
Lcc |
|---|---|---|---|---|---|---|
| **Naive 16-bit** | 23.39 | 46.39 | 29.91 | 74.50 | 4.50 | 57.11 |
| **KIVI 4-bit** | | | | | | |
| Dense | 23.60 | 46.39 | 29.84 | 74.50 | 5.00 | 57.35 |
| K0.5 V0.0 | 23.46 | 46.21 | 28.90 | 74.50 | 5.50 | 56.05 |
| K0.7 V0.0 | 23.35 | 45.40 | 26.46 | 73.50 | 4.83 | 52.41 |
| K0.0 V0.5 | 23.68 | 46.39 | 29.10 | 74.50 | 5.50 | 58.30 |
| K0.0 V0.7 | 24.10 | 45.66 | 27.21 | 74.00 | 5.50 | 59.30 |
| K0.5 V0.5 | 23.22 | 46.06 | 28.18 | 74.00 | 6.00 | 56.04 |
| K0.7 V0.7 | 23.74 | 45.50 | 23.57 | 70.50 | 6.25 | 54.12 |
| **KIVI 2-bit** | | | | | | |
| Dense | 23.33 | 45.47 | 29.69 | 74.50 | 6.50 | 50.38 |
| K0.5 V0.0 | 22.86 | 45.29 | 29.39 | 74.00 | 5.50 | 49.92 |
| K0.7 V0.0 | 22.88 | 44.60 | 26.91 | 73.00 | 4.50 | 43.84 |
| K0.0 V0.5 | 23.65 | 45.67 | 29.05 | 74.00 | 5.50 | 51.94 |
| K0.0 V0.7 | 23.68 | 45.47 | 27.57 | 74.00 | 5.50 | 52.90 |
| K0.5 V0.5 | 22.46 | 45.47 | 28.61 | 74.00 | 4.50 | 48.76 |
| K0.7 V0.7 | 22.72 | 45.18 | 23.84 | 71.00 | 5.12 | 45.68 |

### 4.3 Efficiency Evaluation

A crucial aspect of Mustafar is to ensure that the exploitation of sparsity for compressing the KV cache does not deter the inference latency. Mustafar compensates the overhead of runtime pruning and compression by achieving speedup in the memory-bound SpMV. Figure 6a compares the normalized latency of dense batched MV of cuBLAS with the components of Mustafar sparse attention kernel (Figure 5b): batched SpMV, dense batched MV of local window, runtime pruning, and compression, for input sequence length 2048 for Llama-2 and 4096 for Llama-3 and generation length 1024. In the multi-head attention of Llama-2-7B, pruning introduces 1.84%, compression introduces 6.25%, and MV of local window introduces 0.62% of the cuBLAS execution time in dense inference. In both 50% and 70%, the speedup gained from SpMV kernel more than compensates for the introduced overheads. In 50% sparsity, SpMV takes 81.07% of cuBLAS execution time and for 70% sparsity, SpMV takes 61.87% of cuBLAS execution time. In Grouped-Query Attention of Llama-3-8B, where there is reduced set of KV cache, compression and pruning overhead reduce down to 1.47% and 0.47% of cuBLAS execution time respectively.

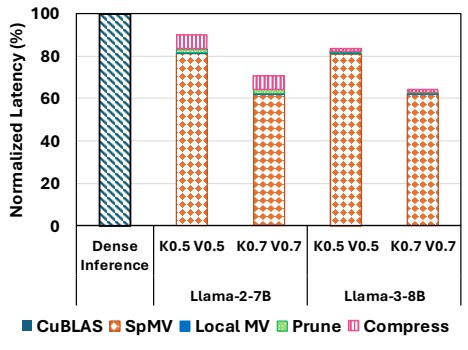

(a) Normalized kernel latency breakdown.

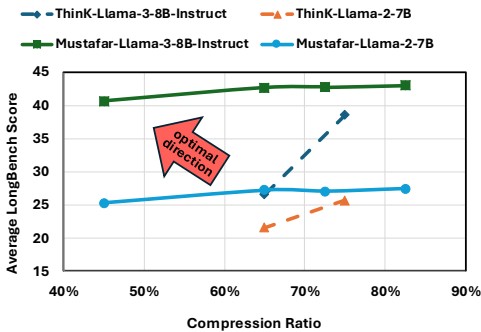

(b) Compression ratio-accuracy comparison of Mustafar and ThinK.

Figure 6: Efficiency evaluation of Mustafar. In (b), compression ratio refers to percentage of compressed size compared to dense KV cache.

Figure 6b compares the KV cache compression ratio (% of size in memory compared to dense KV cache) of Mustafar and ThinK along with the LongBench average score achieved with Llama-2-7B and Llama-3-8B-Instruct. In this plot, the red arrow points to the optimal direction, where a model achieves higher LongBench score while achieving high compression of the KV cache. For ThinK [44] which prune only Key cache, 50% sparsity leads to 75% compression ratio to dense KV cache, and 70% Key cache sparsity leads to 65% compression ratio. In the case of Mustafar where both Key and Value Cache can be pruned, KV cache 50% sparsity leads to 65% compression ratio. The reason behind 15% additional memory footprint is due to the tile offset overhead as shown in Figure 5a and the multiples-of-8 padding enforced to coalesce memory access in GPU. KV cache 70% sparsity leads to 45% compression ratio, 50% sparsity to either Key or Value cache leads to 83% compression ratio, and single-cache 70% sparsity leads to 72.5% compression ratio. Overall, we see that Mustafar is able to achieve better accuracy given the compression ratio, with the compression ratio-accuracy curve closer to the optimal direction than ThinK.

Figure 7 shows the throughput comparison to inference with dense models. For Llama-2 7B, we used input sequence length of 2048 and generated 2048 tokens. For Llama-3 8B, we use input sequence length of 4096 and generated 4096 tokens. For dense baseline, FlashAttention [6] was used on prefill and decode phase. Overall, we see that Mustafar is able to achieve higher throughput as well as support larger batch size owing to the reduced memory footprint of KV cache. In Llama-3, we see that enabling batch size of 8 leads to $2.23\times$ tokens/sec throughput compared the dense inference of

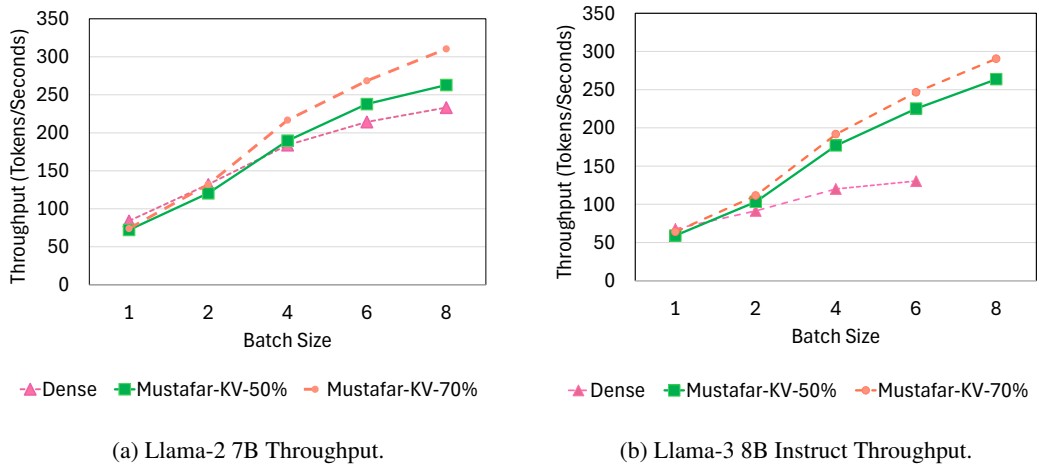

(a) Llama-2 7B Throughput.

(b) Llama-3 8B Instruct Throughput.

Figure 7: Throughput comparison of Mustafar to dense inference.

batch size 6. Even within the same batch size, we see an increased throughput upto $1.89\times$. This is due to the pruning and compression overhead amortized by the speedup of Mustafar sparse attention kernel, leading to faster inference latency. However in batch size 1, we see that throughput is lower than dense inference. This is due to the underutilization of GPU in Mustafar sparse attention kernel with small batch size, where the number of threadblocks is smaller than the number of SMs. We provide additional throughput comparison with different input:output token ratios on Appendix C.3.

## 5  Related Work

**KV cache compression** Alongside aforementioned work in KV cache pruning [44, 31], quantization [30, 15, 48, 52], token-wise eviction [51, 29, 25, 8, 1, 11], and low-rank approximation [47, 4, 37, 50, 26], KV cache offloading [24, 28, 13, 5] evicts KV cache to CPU memory and speculatively prefetchs critical tokens' KV cache. Layer-centric compression [49, 27] applies different level of compression to different layers, adhering to layer-wise importance. Head-level compression [10, 39] applies different level of compression to each heads, from the observation that not all heads contribute equally. Phase-specific compression [42] applies different strategy for prefill and decode phase, with information retention prioritized in prefill and heavy hitter selection applied on decode phase.

**System/Kernel for Attention** While Mustafar attention kernel focuses on operating directly on the bitmap-compressed sparse KV cache, there exists various contributions from the system and kernel-levels to optimize for attention. PagedAttention [23] introduces an paging-inspired attention algorithm that partitions KV cache into memory blocks to reduce memory fragmentation and efficient sharing across sequences. FlashDecoding[16] introduces double-buffering to accelerate memory-bound GeMM of decode phase. FlashInfer [45] unifies KV cache format using a block-sparse representation for an efficient management of KV cache that leads to increased throughput. Loki [36] uses a sparse attention method that leverages the low-dimensionality of key vectors to perform an approximate attention in a reduced PCA space.

## 6  Conclusion and Limitations

In this work, we demonstrate that unstructured sparsity presents a powerful and novel solution for KV cache pruning. By removing constraints on the pruning pattern, we show that per-token magnitude-based pruning achieves high sparsity while maintaining model accuracy. To unlock the practical benefits of unstructured sparsity, we introduce a bitmap-based sparse format and a custom attention kernel that directly operates on compressed KV cache. Together, our pruning strategy, sparse format, and custom kernel form an end-to-end system that substantially reduces KV cache memory usage and improves throughput, making it possible to support longer contexts and more efficient inference. Mustafar establishes a foundation for future efforts to integrate unstructured sparsity into practical LLM deployment pipelines and opens new directions for memory-efficient LLM inference at scale. In future work, we plan to explore the joint effect of leveraging KV sparsity of Mustafar with sparsity in weights derived by works such as output-aware weight pruning [38], pruning with low-rank adapters for accuracy retention [34, 33], and activation-aware calibration and efficiency enhancement [32, 46]. Additionally, this paper focuses on showing that unstructure sparsity can prune both Key and Value caches to a higher sparsity with better accuracy than structured sparsity, leaving our method's ability to map arbitrary sparsity degree untouched. While we explore higher sparsity uniformly applied to the entire KV cache in Appendix A.4, a future work involves deriving the optimal target sparsity to a smaller granularity (e.g. per-head or per-layer) to maximize sparsity and accuracy retention.

## Acknowledgments and Disclosure of Funding

We gratefully acknowledge the support of National Science Foundation (NSF) under program PPoSS, Award Number 2316177.

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

# A  Extended Evaluation

## A.1  Section 2 Methodology Applied to LLaMA-2 7B

We follow the same methodology of exploring pruning direction and output-awareness on Llama-2-7B to further solidify our findings on a model architecture with Multi-Head Attention. In Table 7, we observe a similar trend to that of Llama-3-8B-Instruct in Section 2. Unstructured pruning outperforms structured pruning of ThinK [44], with ouput-awareness bringing a small accuracy increase to pure magnitude-based pruning.

Table 7: Comparison of ThinK [44] structured pruning, per-token magnitude-based unstructured pruning, and per-token output-aware unstructured pruning on LongBench [3] with Llama-2-7B Key cache.

| Task | Dense | $K_s = 0.5$ | | | $K_s = 0.7$ | | |
| --- | --- | --- | --- | --- | --- | --- | --- |
| | | ThinK (Structured) | Unstructured Output-aware | Unstructured Magnitude | ThinK (Structured) | Unstructured Output-aware | Unstructured Magnitude |
| Average | 27.51 | 25.70 | **27.55** | 27.46 | 21.57 | **26.78** | 26.17 |
| SingleDoc QA | 15.53 | 16.28 | **15.52** | 15.37 | 14.17 | **15.82** | 14.43 |
| MultiDoc QA | 7.10 | 6.30 | 6.90 | **7.23** | 4.33 | 6.44 | **6.62** |
| Summarization | 14.02 | 7.46 | **14.51** | 13.91 | 9.28 | **12.99** | 11.64 |
| Few-shot | 65.13 | 64.34 | **65.20** | 65.00 | 57.81 | **63.77** | 63.65 |
| Synthetic | 4.17 | 2.42 | **3.98** | 3.90 | 4.35 | 3.00 | **3.08** |
| Code | 63.24 | 61.57 | 63.22 | **63.52** | 39.85 | **62.67** | 61.73 |

In Table 8, a unique phenomenon is the stark contrast of model accuracy in per-channel unstructured pruning methods. Whereas per-channel magnitude-based pruning of Table 2 show good model accuracy retention for Llama-3-8B-Instruct, for Llama-2-7B we see that accuracy degradation is very severe. Nevertheless, concurrent to our previous finding, we once again see that per-channel pruning achieves the same level of accuracy retention to per-token pruning as output-awareness is applied. This highlights the importance of output-awareness in Value cache pruning. In Table 9 we see that the model accuracy of 70% unstructured sparsity on both Key and Value cache achieves similar accuracy to 50% ThinK pruning.

Table 8: Comparison of ThinK [44] structured pruning, per-channel magnitude-based unstructured pruning, per-channel output-aware unstructured pruning, and per-token magnitude-based pruning on LongBench [3] with Llama-2-7B Value cache.

| Task | Dense | $V_s = 0.5$ | | | | $V_s = 0.7$ | | | |
| --- | --- | --- | --- | --- | --- | --- | --- | --- | --- |
| | | ThinK (Structured) | Magnitude (Per-channel) | Output-aware (Per-channel) | Magnitude (Per-token) | ThinK (Structured) | Magnitude (Per-channel) | Output-aware (Per-channel) | Magnitude (Per-token) |
| Average | 27.51 | 24.59 | 6.16 | 27.33 | **27.39** | 21.10 | 5.81 | 26.30 | **27.05** |
| SingleDoc QA | 15.53 | 12.64 | 1.68 | **15.96** | 15.62 | 10.05 | 1.60 | **15.48** | 15.17 |
| MultiDoc QA | 7.10 | 7.37 | 2.17 | **6.97** | 6.92 | 7.15 | 1.82 | 6.97 | **7.10** |
| Summarization | 14.02 | 9.18 | 4.51 | **13.98** | 13.54 | 9.10 | 3.15 | **13.06** | 12.67 |
| Few-shot | 65.13 | 61.82 | 9.93 | 64.07 | **64.92** | 57.12 | 8.83 | 60.09 | **64.80** |
| Synthetic | 4.17 | 3.86 | 1.82 | **4.45** | 4.20 | 1.65 | 2.45 | **4.69** | 3.95 |
| Code | 63.24 | 56.31 | 20.03 | 62.72 | **63.44** | 41.96 | 20.90 | 62.34 | **62.85** |

Table 9: Longbench evaluation of Llama-2 7B with KV cache per-token magnitude-based pruning

| Task | Llama-2-7B | | |
| --- | --- | --- | --- |
| | Dense | $K_s = 0.5$ $V_s = 0.5$ | $K_s = 0.7$ $V_s = 0.7$ |
| Average | 27.51 | 27.23 | 24.71 |
| SingleDoc QA | 15.53 | 15.21 | 13.62 |
| MultiDoc QA | 7.10 | 7.11 | 6.78 |
| Summarization | 14.02 | 13.61 | 6.84 |
| Few-shot | 65.13 | 64.84 | 62.59 |
| Synthetic | 4.17 | 3.55 | 2.63 |
| Code | 63.24 | 63.16 | 60.35 |

## A.2 Scaling to Larger Model

In Table 10, we include the accuracy evaluation of Mustafar per-token magnitude-based pruning on Llama-2-13B-chat [40], validating the effectiveness of Mustafar on model with larger size. While unstructured pruning constantly outperforms structured sparsity, we see that the Key cache of Llama-2-13B-chat is more susceptible to accuracy degradation at 70% sparsity (yellow). In this case, we leverage the modularity of Mustafar, being able to apply different target sparsity to Key and Value cache to find the best combination, to use 50% sparsity for Key Cache and 70% sparsity for Value cache (pink), thereby reaching the higher overall sparsity while maintaining the model accuracy.

Table 10: Mustafar accuracy with Llama-2-13B-chat on LongBench

| KV Sparsity | Single-Document QA | | | Multi-Document QA | | | Summarization | | | Few-shot Learning | | | Synthetic | | Code | | Avg. |
|---|---|---|---|---|---|---|---|---|---|---|---|---|---|---|---|---|---|
| | NrtvQA | Qasper | MF-en | HotpotQA | 2WikiMQA | Musique | GovReport | QMSum | MultiNews | TREC | TriviaQA | SAMSum | PCount | PRe | Lec | RBP | |
| | | | | | | | | | *Llama-2-13B-Chat* | | | | | | | | |
| Dense | 18.54 | 24.09 | 37.01 | 36.43 | 31.40 | 15.81 | 24.48 | 20.25 | 25.74 | 67.50 | 86.90 | 42.07 | 3.00 | 12.00 | 50.12 | 50.53 | **34.12** |
| ThinK0.5 | 16.95 | 22.39 | 37.54 | 34.00 | 29.93 | 14.33 | 24.49 | 20.21 | 24.78 | 67.50 | 87.16 | 40.53 | 2.55 | 13.07 | 45.79 | 46.23 | **32.80** |
| K0.5 V0.0 | 18.46 | 23.12 | 37.26 | 37.16 | 31.18 | 15.56 | 23.90 | 20.55 | 25.57 | 67.50 | 87.23 | 41.99 | 3.00 | 11.50 | 50.33 | 48.88 | **33.95** |
| ThinK0.7 | 17.86 | 19.93 | 32.37 | 33.03 | 27.22 | 13.99 | 21.19 | 19.47 | 12.04 | 59.0 | 86.67 | 31.26 | 1.54 | 1.87 | 27.79 | 29.35 | **27.16** |
| K0.7 V0.0 | 14.63 | 20.97 | 34.05 | 34.70 | 30.69 | 13.72 | 10.60 | 20.01 | 7.63 | 61.00 | 81.91 | 37.76 | 1.00 | 1.00 | 45.29 | 33.54 | **28.03** |
| K0.0 V0.5 | 18.75 | 23.68 | 37.34 | 36.83 | 31.36 | 15.50 | 23.97 | 20.83 | 25.46 | 67.50 | 87.20 | 41.45 | 2.50 | 10.00 | 49.32 | 49.37 | **33.82** |
| K0.0 V0.7 | 19.29 | 22.90 | 37.65 | 36.57 | 31.24 | 15.35 | 22.44 | 20.52 | 24.75 | 68.00 | 87.49 | 40.55 | 2.50 | 8.10 | 49.33 | 49.14 | **33.49** |
| K0.5 V0.5 | 19.08 | 22.66 | 36.97 | 37.25 | 31.38 | 15.46 | 23.70 | 20.66 | 25.39 | 67.50 | 87.23 | 40.59 | 3.00 | 10.10 | 49.39 | 48.06 | **33.64** |
| K0.5 V0.7 | 18.60 | 22.57 | 37.18 | 35.40 | 31.55 | 15.25 | 22.30 | 20.43 | 24.81 | 68.00 | 87.23 | 39.91 | 2.50 | 7.70 | 49.02 | 47.38 | **33.24** |
| K0.7 V0.7 | 17.86 | 19.93 | 32.37 | 33.03 | 27.22 | 13.99 | 21.19 | 19.47 | 12.04 | 59.00 | 86.67 | 31.26 | 1.54 | 1.87 | 27.79 | 29.35 | **27.16** |

## A.3 Evaluation on RULER

For a more diverse evaluation, we evaluate Llama-3.1-8B-Instruct on RULER [17] benchmark for context length of 65,536 tokens.

Table 11: Accuracy comparison on RULER benchmark

| Sparsity | Method | Needle-Single1 | Needle-Single2 | Needle-MultiKey1 | Needle-MultiKey2 | Needle-MultiQuery | Needle-MultiValue | QA-1 | QA-2 | Variable Tracking | Freq. Words Extract. |
|---|---|---|---|---|---|---|---|---|---|---|---|
| | | | | | | *Llama-3.1-8B-Instruct* | | | | | |
| Dense | — | 1.000 | 1.000 | 0.990 | 0.979 | 0.990 | 0.979 | 0.844 | 0.594 | 0.973 | 0.851 |
| Key 50% | ThinK | 1.000 | 1.000 | 0.990 | 0.979 | 0.995 | 0.969 | 0.833 | 0.594 | 0.919 | 0.854 |
| | Mustafar | 1.000 | 1.000 | 0.990 | 0.979 | 0.995 | 0.996 | 0.833 | 0.573 | 0.971 | 0.813 |
| Key 70% | ThinK | 0.448 | 0.490 | 0.229 | 0.188 | 0.646 | 0.487 | 0.615 | 0.510 | 0.208 | 0.427 |
| | Mustafar | 1.000 | 1.000 | 0.990 | 0.969 | 0.992 | 0.903 | 0.833 | 0.594 | 0.966 | 0.823 |
| Value 50% | ThinK | 1.000 | 1.000 | 0.990 | 0.969 | 0.914 | 0.958 | 0.823 | 0.573 | 0.910 | 0.792 |
| | Mustafar | 1.000 | 1.000 | 0.979 | 0.995 | 0.995 | 0.971 | 0.833 | 0.604 | 0.983 | 0.830 |
| Value 70% | ThinK | 0.948 | 0.927 | 0.948 | 0.510 | 0.698 | 0.688 | 0.646 | 0.500 | 0.558 | 0.677 |
| | Mustafar | 1.000 | 1.000 | 1.000 | 0.979 | 0.992 | 0.969 | 0.833 | 0.594 | 0.985 | 0.826 |
| Key&Value 50% | ThinK | 0.958 | 1.000 | 0.948 | 0.854 | 0.828 | 0.956 | 0.740 | 0.531 | 0.742 | 0.823 |
| | Mustafar | 1.000 | 1.000 | 0.990 | 0.979 | 0.997 | 0.997 | 0.833 | 0.573 | 0.862 | 0.809 |
| Key&Value 70% | ThinK | 0.000 | 0.073 | 0.000 | 0.000 | 0.000 | 0.000 | 0.219 | 0.250 | 0.000 | 0.035 |
| | Mustafar | 1.000 | 1.000 | 0.990 | 0.969 | 0.995 | 0.914 | 0.833 | 0.583 | 0.869 | 0.799 |

As shown in Table 11, even in the challenging Needle-in-a-Haystack scenarios with multiple keys and queries, Mustafar maintains accuracy comparable to the dense model. It also outperforms the structured pruning baseline ThinK, with particularly notable gains at 70% joint Key-Value sparsity. While structured pruning does perform well in isolated cases, such as the Needle-Single tasks for 70% Value sparsity, it exhibits significant accuracy drops in other tasks. In contrast, Mustafar's unstructured sparsity consistently preserves accuracy across all tasks. This contrast highlights the versatility of unstructured sparsity in adapting to diverse task requirements.

## A.4 Higher Sparsity

While the main paper primarily focused on 50% and 70% sparsity of both Key and Value Cache, we present the performance of Mustafar per-token magnitude-based pruning of KV cache 80% and 90% sparsity in Table 12. While we see that Key cache suffers from accuracy degradation in higher sparsity, Value cache, despite the even distribution of element magnitude as in Figure 2b, retains some level of the model accuracy even at 90% sparsity on selective tasks. Model accuracy is retained for tasks such as 2WikiMultihopQA (pink), while degraded significantly in tasks such as GovReport (yellow).

Table 12: Mustafar accuracy with Llama-3-8B-Instruct on LongBench

| KV Sparsity | Single-Document QA | | | Multi-Document QA | | | Summarization | | | Few-shot Learning | | | Synthetic | | Code | | Avg. |
| --- | --- | --- | --- | --- | --- | --- | --- | --- | --- | --- | --- | --- | --- | --- | --- | --- | --- |
| | NrtvQA | Qasper | MF-en | HotpotQA | 2WikiMQA | Musique | GovReport | QMSum | MultiNews | TREC | TriviaQA | SAMSum | PCount | PRe | Lcc | RBP | |
| Llama-3-8B-Instruct | | | | | | | | | | | | | | | | | |
| Dense | 23.39 | 43.38 | 43.22 | 46.39 | 38.66 | 23.22 | 29.91 | 22.56 | 27.77 | 74.5 | 90.28 | 42.11 | 4.50 | 70.00 | 57.11 | 54.05 | **43.19** |
| K0.8 V0.0 | 22.67 | 39.08 | 39.44 | 44.98 | 38.51 | 21.94 | 21.75 | 21.00 | 23.88 | 69.00 | 90.24 | 36.92 | 7.50 | 64.50 | 49.15 | 45.79 | **39.77** |
| K0.9 V0.0 | 19.90 | 28.92 | 35.21 | 41.56 | 30.77 | 18.89 | 11.78 | 18.40 | 14.95 | 39.50 | 81.79 | 29.18 | 2.75 | 61.50 | 40.30 | 33.46 | **31.80** |
| K0.0 V0.8 | 24.48 | 42.54 | 43.96 | 45.48 | 38.71 | 22.46 | 24.47 | 21.64 | 25.09 | 73.00 | 90.11 | 39.03 | 5.62 | 64.00 | 56.39 | 56.54 | **42.22** |
| K0.0 V0.9 | 24.12 | 37.90 | 42.53 | 44.68 | 38.29 | 21.99 | 20.22 | 21.29 | 21.61 | 69.00 | 90.15 | 36.04 | 3.29 | 62.50 | 55.87 | 53.59 | **40.19** |
| K0.8, V0.8 | 21.82 | 36.53 | 38.61 | 44.38 | 36.31 | 21.33 | 19.18 | 20.74 | 20.80 | 59.50 | 88.27 | 32.68 | 5.25 | 64.00 | 51.03 | 48.29 | **38.05** |
| K0.9, V0.9 | 17.47 | 24.13 | 30.64 | 38.63 | 29.24 | 17.24 | 13.50 | 19.67 | 15.03 | 35.50 | 75.29 | 27.39 | 5.50 | 63.00 | 41.77 | 34.39 | **30.52** |

## B Comparison with Semi-structured Sparsity

Between the structured pruning of rows and columns, and unstructured pruning of element, lies the 2:4 semi-structured sparsity where 2 out of 4 consecutive elements are non-zero, enforcing a global 50% sparsity. Supported by NVIDIA Sparse Tensor Cores, 2:4 semi-structured sparsity also pursue the same objectives of Mustafar bitmap-based sparse format (Figure 5a), maximal compression and fast computation. In Table 13, we apply 2:4 semi-structured pruning to the per-token magnitude-based scheme. Comparing semi-structured sparsity to Key, Value, and both Key and Value cache to unstructured sparsity of Mustafar, we see that unstructured sparsity constantly outperforms semi-structured pattern of the same sparsity. This emphasizes the impact of fine-grained unstructured sparsity of element-wise pruning in model accuracy retention.

Table 13: Comparison of 2:4 semi-structured and unstructured sparsity with Llama-3-8B-Instruct on LongBench

| KV Sparsity | Single-Document QA | | | Multi-Document QA | | | Summarization | | | Few-shot Learning | | | Synthetic | | Code | | Avg. |
| --- | --- | --- | --- | --- | --- | --- | --- | --- | --- | --- | --- | --- | --- | --- | --- | --- | --- |
| | NrtvQA | Qasper | MF-en | HotpotQA | 2WikiMQA | Musique | GovReport | QMSum | MultiNews | TREC | TriviaQA | SAMSum | PCount | PRe | Lcc | RBP | |
| Llama-3-8B-Instruct | | | | | | | | | | | | | | | | | |
| Dense | 23.39 | 43.38 | 43.22 | 46.39 | 38.66 | 23.22 | 29.91 | 22.56 | 27.77 | 74.5 | 90.28 | 42.11 | 4.50 | 70.00 | 57.11 | 54.05 | **43.19** |
| K0.5 (2:4) | 21.79 | 39.77 | 42.34 | 45.15 | 38.81 | 21.72 | 24.34 | 22.21 | 25.44 | 69.50 | 90.87 | 39.10 | 7.00 | 62.50 | 54.33 | 50.29 | **40.95** |
| K0.5 (Unstructured) | 23.40 | 43.68 | 43.63 | 46.00 | 38.60 | 22.72 | 29.39 | 22.33 | 27.64 | 74.50 | 90.66 | 41.09 | 5.00 | 68.50 | 55.89 | 52.39 | **42.84** |
| V0.5 (2:4) | 23.69 | 42.72 | 43.94 | 45.48 | 39.42 | 22.78 | 28.51 | 22.53 | 26.66 | 73.50 | 90.31 | 40.92 | 4.50 | 68.00 | 58.35 | 55.68 | **42.94** |
| V0.5 (Unstructured) | 23.80 | 43.14 | 43.32 | 46.28 | 39.42 | 22.97 | 29.18 | 22.70 | 27.13 | 74.50 | 90.50 | 41.74 | 5.00 | 67.50 | 57.23 | 54.30 | **43.04** |
| K0.5(2:4) V0.5(2:4) | 22.32 | 39.42 | 42.64 | 45.45 | 38.25 | 21.52 | 23.41 | 21.82 | 24.38 | 69 | 91.04 | 39.59 | 7.5 | 62.5 | 55.02 | 50.41 | **40.89** |
| K0.5 V0.5 (Unstructured) | 23.40 | 46.63 | 42.98 | 46.28 | 39.27 | 23.13 | 28.29 | 22.78 | 27.07 | 74.00 | 90.58 | 39.97 | 5.00 | 67.00 | 55.54 | 53.46 | **42.65** |

## C  Sparse Attention Kernel Details

As a supplement to Section 3, we offer more detail onto the Mustafar sparse attention kernel, which accelerates memory-bound batch SpMV.

### C.1  Load-as-Compressed, Compute-as-Dense Pipeline

Crucial insight of accelerating SpMV involves reducing the data movement between the GPU global memory and the local memory of each GPU Streaming Multiprocessor. First proposed by FlashLLM [43], load-as-compressed, compute-as-dense pipeline as shown in Figure 8 involves sending each matrix tile in the corresponding compressed form to the SM registers ('gmem2reg' in the figure), decompressing the compressed tile into the dense from to the shared memory ('extract'), then initializing computation on the next pipeline stage ('smem2tc'). Computation is mapped to tensor core to utilize

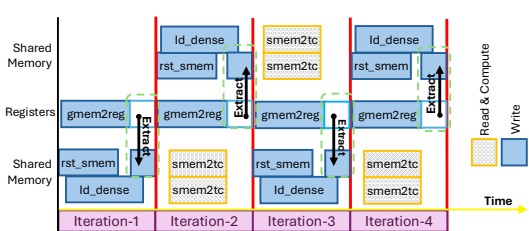

Figure 8: Load-as-compressed, compute-as-compute pipeline of FlashLLM [43]

the high fp16 compute throughput. To map MV, unused N dimensions are padded to zero for computation. Non-zero thread-tile of $1 \times 64$ in Figure 5a represents the granularity of non-zeros that a warp thread decompresses at a pipeline stage. Each warp thread decompresses 2 thread-tile per stage using the corresponding bitmap to determine the correct position of each non-zero. Effectively, each warp operates on a $64 \times 64$ sized matrix tile at a time.

### C.2  KV Cache Management

Tile size of $64 \times 64$ of each warp-tile (pink tiles in Figure 9), requires the KV cache to be compressed and appended to the existing KV cache in token groups of 64. Due to the dynamic nature of KV cache where new entries are added during generation, a kernel-compatible management of KV cache update is necessary. That is, (1) column tiling direction of KV cache must be orthogonal to the dimension that is being multiplied with: Key cache is multiplied on the channel-dimension, thus column tiling is across token dimension (yellow arrow in Figure 9a), value cache is multiplied on the token-dimension, thus column-tiling is across the channel dimension (yellow arrow in Figure 9b).

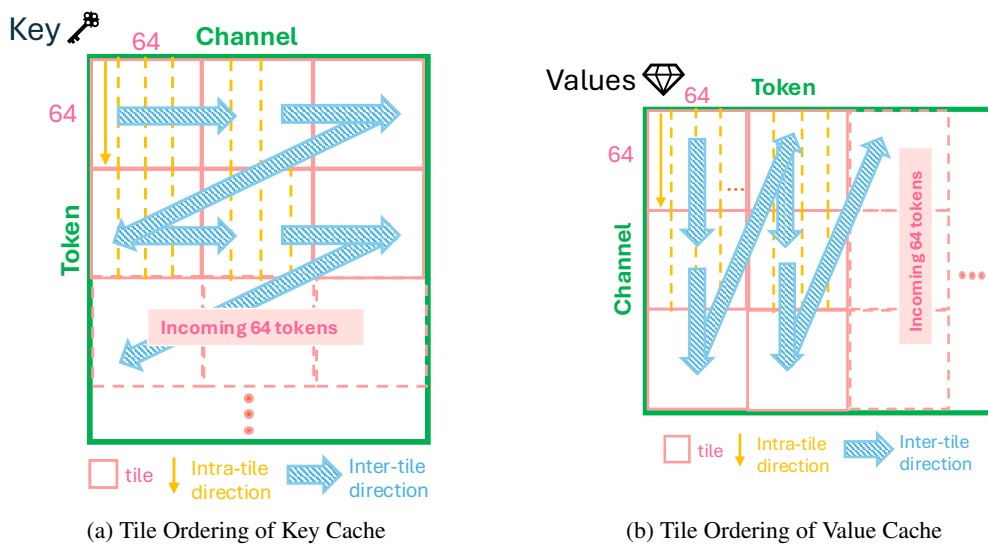

(a) Tile Ordering of Key Cache                    (b) Tile Ordering of Value Cache

Figure 9: Tile ordering scheme of Key and Value cache

(2), the layout of warp-tile must ensure that newly compressed tokens' KV cache can be appended to the existing compressed KV cache. As newly compressed KV cache are added onto the token

dimension, traversal across multiple warp-tiles is done along channel-major dimension for both Key and and Value caches so that the compressed KV cache of the new tokens can be appended at the end.

## C.3 Decode Speed Evaluation

Extrapolating on Figure 7, we evaluate Mustafar decoding on various input:output token ratios with batch size 4. For Llama-2-7B, we use input sequence length of 2048. For Llama-3-8B-Instruct, we use input sequence length of 4096. We use output sequence lengths of 512, 1024, and 2048.

Table 14: Decode speed comparison with dense inference

| Model | KV Format | TTFT | Decode Speed (decode 512) | Decode Speed (decode 1024) | Decode Speed (decode 2048) |
|---|---|---|---|---|---|
| Llama2 | Dense | 1.396 sec | 88.685 tokens / sec | 88.512 tokens / sec | 79.185 tokens / sec |
| | Mustafar K0.5 V0.5 | 2.532 sec | 89.452 tokens / sec | 89.514 tokens / sec | 85.687 tokens / sec |
| | Mustafar K0.7 V0.7 | 2.249 sec | 96.386 tokens / sec | 97.436 tokens / sec | 95.120 tokens / sec |
| Llama3 | Dense | 2.769 sec | 61.993 tokens / sec | 61.220 tokens / sec | 59.242 tokens / sec |
| | Mustafar K0.5 V0.5 | 3.269 sec | 78.434 tokens / sec | 83.768 tokens / sec | 83.303 tokens / sec |
| | Mustafar K0.7 V0.7 | 3.151 sec | 84.065 tokens / sec | 88.293 tokens / sec | 89.699 tokens / sec |

While Figure 7 measured the token throughput by considering both input and output tokens processed, in Table 14 we derived the average decoding speed by measuring the end-to-end duration, and dividing it to the number of tokens generated to penalize Mustafar with the overhead of KV cache pruning and compression in both prefill and decode stages.

While time-to-first-token is delayed due to the overhead of pruning and compressing the KV cache during the prefill stage, the delay is offset by the accelerated attention computation during decoding, resulting in higher overall token generation throughput. Notably, Llama-3 exhibits a larger performance gain compared to Llama-2, as its GQA architecture reduces the overhead of KV cache pruning and compression.

