# OpenReview forum: "MUSTAFAR: Promoting Unstructured Sparsity for KV Cache Pruning in LLM Inference"
_NeurIPS.cc/2025/Conference — NeurIPS 2025 poster_

### Official Review · Reviewer_QfM2 · 2025-06-17

**Clarity:** 3
**Significance:** 3
**Originality:** 3
**Rating:** 4
**Confidence:** 3

**Summary:**

This paper introduces an unstructured pruning method for KV cache compression. At first, this paper compares ThinK, structured pruning, per-token magnitude-based unstructured pruning, and per-token output-aware unstructured pruning to determine the pruning strategy for the Key and Value cache, respectively. Moreover, a sparse attention kernel is implemented to show that their method is able to provide real efficiency for current LLMs. Basically, this method is faster than previous approaches when the compression rate is the same, while achieving a lower compression rate than previous approaches when the performance is the same.

**Questions:**

1. Does this method consider the difference in importance between layer-wise and head-wise? Could the author show some toy experiments to demonstrate this question?

2. Is this method effective for models using non-standard KV cache architectures such as Mamba and RWKV?

**Ethical Concerns:**

["NO or VERY MINOR ethics concerns only"]

**Final Justification:**

I recommend accepting this paper (Borderline accept).

**Limitations:**

Yes.

**Paper Formatting Concerns:**

No concerns.

**Quality:**

3

**Strengths And Weaknesses:**

Strengths

1. The overall logic is clear. The proposed motivation experiments clearly support their method.

2. The evaluation is sufficient on various datasets.

3. The implementation of a real sparse kernel makes their method valuable in the real world.


Weaknesses

1. Is the bitmap format suitable for low sparsity scenarios? For example, at a sparsity rate <30%, the cost of bitmap+index may be greater than that of dense representation? Could the author provide some examples for this?

2. This paper only introduces ThinK as a structured baseline in method comparison. Could the author add more baseline methods into the comparison? If not possible, I also hope the author can explain why.

3. Typos: Miss '.' in the caption of Table 1. Line 101, 103, 110: 'Query vector(blue) ' -> 'Query vector (blue)', 'pruning score(green)' -> 'pruning score (green)', and 'Group Query Attention(GQA) ' -> 'Group Query Attention (GQA)'.

---

> ### Author Rebuttal · Authors · 2025-07-30
>
> We thank the reviewer for the constructive feedback.
>
> > **W1. Is the bitmap format suitable for low sparsity scenarios? For example, at a sparsity rate <30%, the cost of bitmap+index may be greater than that of dense representation? Could the author provide some examples for this?**
>
> Our bitmap format compresses the KV cache using a per-element bitmap, a per-64-element tile offset (int32), and the compressed non-zero values. For a 4096 (tokens) × 4096 (channels across all attention heads) FP16 matrix, this corresponds to a fixed 2 MB bitmap, a 1 MB tile offset, and a variable-sized compressed non-zero array scaled by the sparsity (i.e., 2 × 4096 × 4096 × sparsity bytes). Given that the original dense matrix occupies 33 MB, compression yields a size benefit when sparsity exceeds 9.4%.
>
> However, while our format primarily targets memory compression, we also optimize for latency. Our attention kernel directly leverages the bitmap format to accelerate the memory-bound attention computation, where kernel performance scales with the compression ratio to impose a more stringent requirement on sparsity.
>
> In terms of raw kernel throughput, excluding runtime pruning and compression overhead, our method surpasses cuBLAS at 30% sparsity and outperforms FlashAttention during decoding at 40% sparsity.
>
> > **W2. This paper only introduces ThinK as a structured baseline in method comparison. Could the author add more baseline methods into the comparison? If not possible, I also hope the author can explain why.**
>
> We selected ThinK as the sole baseline because, to the best of our knowledge, it is the only existing work that performs channel-wise (i.e., per-token) pruning, which is compatible with and can be jointly applied alongside orthogonal KV cache compression techniques.
>
> In our paper, what we referred to as “token-wise eviction” corresponds to structured pruning along the channel dimension. We view this form of structured pruning (H2O, SnapKV [1], AdaKV [2], and HeadKV [3]) not as a direct competitor, but rather as a complementary technique that Mustafar can be layered on top of.
>
> In Section 4.2, we evaluated the effect of joint application with token-wise eviction of H2O to demonstrate this complementarity. Specifically, for the remaining KV cache after such token-wise pruning, Mustafar can be applied to further prune the cache at a finer, per-token granularity. We additionally demonstrate the effect of joint usage with HeadKV, a head-granular token-wise pruning method, in our response to Reviewer PxDY, W1.
>
> > **W3. Typos: Miss '.' in the caption of Table 1. Line 101, 103, 110: 'Query vector(blue)' → 'Query vector (blue)', 'pruning score(green)' → 'pruning score (green)', and 'Group Query Attention(GQA)' → 'Group Query Attention (GQA)'.**
>
> Thank you for pointing this out. We will fix the typos in the revised version.
>
> > **Q1. Does this method consider the difference in importance between layer-wise and head-wise? Could the author show some toy experiments to demonstrate this question?**
>
> Our evaluation focused on the ability to achieve higher sparsity using unstructured sparsity compared to structured sparsity, using a globally uniform sparsity level of 50% and 70% applied across a model's entire KV cache. While this setup was noted in Section 6, and we explicitly pointed to layer-wise (as explored in What Matters in Transformers? [4]) and head-wise (as explored in HeadKV) sparsity as future directions, we acknowledge the reviewer’s comment and have conducted a toy experiment to explore this further.
>
> We first combine Mustafar's 60% KV cache pruning with HeadKV, a head-level, token-wise eviction method, to reduce experiment time. This setting is denoted as HeadKV + Mustafar (uniform 60%). Next, we use the head-level importance score from HeadKV to assign 50% sparsity to the more important half of each layer's heads, and 70% sparsity to the less important half, averaging to an effective 60% sparsity overall. This setting is denoted as HeadKV + Mustafar (head-wise). We evaluate this setup on one representative task from each LongBench category: single-document QA (NarrativeQA), multi-document QA (HotpotQA), summarization (GovReport), few-shot learning (TREC), synthetic (Pcount), and code (LCC), on Llama-3-8B-Instruct.
>
> | Method                                | narrativeQA | hotpotQA | gov_report | TREC  | Pcount | LCC   |
> |---------------------------------------|-------------|----------|------------|-------|--------|-------|
> | HeadKV [KV size = 128] Dense                    | 22.33       | 43.66    | 21.67      | 72.0  | 5.67   | 59.39 |
> | HeadKV + Mustafar (uniform 60%)      | 22.14       | 41.85    | 20.57      | 70.50 | 5.87   | 56.22 |
> | HeadKV + Mustafar (head-wise)    | 23.34       | 41.99    | 20.61      | 71.0  | 5.71   | 57.46 |
>
>
> The table above shows that, in most tasks, except for a slight deviation on Pcount, varying head-level sparsity based on head-level importance scores yields better model accuracy than using uniform sparsity. While our current setup only applies either 50% or 70% sparsity, we note that Mustafar's sparse format and attention kernel support arbitrary sparsity values. This flexibility enables more precise alignment between head-level importance and sparsity, which we expect could further improve accuracy preservation.
>
> > **Q2. Is this method effective for models using non-standard KV cache architectures such as Mamba and RWKV?**
>
> Mustafar KV cache compression method is primarily designed to address the growing memory footprint associated with conventional attention-based models, where the KV cache scales linearly with the context length. As context windows grow, so does the memory cost, making compression both necessary and impactful.
>
> In contrast, non-standard architectures like Mamba and RWKV, not only less common in production compared to conventional attention-based models, employ constant-sized compressive memory, which decouples memory usage from context length. As such, they do not face the same scalability challenges and are not the primary target of our method.
>
> References:
> [1] Yuhong Li, Yingbing Huang, Bowen Yang, Bharat Venkitesh, Acyr Locatelli, Hanchen Ye, Tianle Cai, Patrick Lewis, Deming Chen. *SnapKV: LLM Knows What You are Looking for Before Generation.*
>
> [2] Yuan Feng, Junlin Lv, Yukun Cao, Xike Xie, S. Kevin Zhou. *Ada-KV: Optimizing KV Cache Eviction by Adaptive Budget Allocation for Efficient LLM Inference.*
>
> [3] Yu Fu, Zefan Cai, Abedelkadir Asi, Wayne Xiong, Yue Dong, Wen Xiao. *Not All Heads Matter: A Head-Level KV Cache Compression Method with Integrated Retrieval and Reasoning.*
>
> [4] Shwai He, Guoheng Sun, Zheyu Shen, Ang Li. *What Matters in Transformers? Not All Attention is Needed.*

---

> > ### Comment · Reviewer_QfM2 · 2025-08-03
> >
> > Good paper! I recommend accepting this paper.

---

### Official Review · Reviewer_PxDY · 2025-06-30

**Clarity:** 3
**Significance:** 2
**Originality:** 2
**Rating:** 4
**Confidence:** 4

**Summary:**

This paper introduces the MUSTAFAR method, which optimizes key-value cache pruning in large language model inference through unstructured sparsity to address the issue of excessive memory overhead of KV Cache in long-context scenarios. The authors claim that compared to traditional structured pruning methods, it can maintain model accuracy at higher sparsity rates. Specifically, the Key Cache employs element-magnitude-based per-token pruning combined with an output-aware strategy, while the Value Cache can be efficiently compressed through simple per-token magnitude pruning. To effectively utilize sparsity, the paper designs a bitmap-based sparse representation and customized CUDA attention kernels that directly compute on the pruned KV Cache, significantly reducing memory bandwidth requirements and accelerating attention operations.

**Questions:**

Please refer to Strengths And Weaknesses for details..

**Ethical Concerns:**

["NO or VERY MINOR ethics concerns only"]

**Final Justification:**

This paper presents a practical approach to KV cache compression through unstructured sparsity, achieving significant memory savings while maintaining model accuracy. The proposed method increases TTFT, and the model used is a bit outdated. Overall, I believe the paper is worthy of a borderline accept.

**Limitations:**

yes

**Quality:**

2

**Strengths And Weaknesses:**

### Strengths

S1)This paper presents a practical approach to KV cache compression through unstructured sparsity, achieving significant memory savings while maintaining model accuracy.

S2)The proposed method is intuitive and well-explained, with clear writing that makes the technical content accessible.

### Weaknesses

My primary concerns regarding this paper involve several missing critical experiments:

W1) The baseline comparison is limited to ThinK, while omitting recent highly relevant methods in KV cache compression such as HeadKV [1]. The reported LongBench results appear inferior to HeadKV's performance, particularly since HeadKV demonstrates effectiveness even when retaining less than 5% of KV cache.

W2) The evaluation lacks standard long-context benchmarks like "Needle in a Haystack", which have become prevalent in KV cache compression research [1-3]. LongBench's average sequence length may not adequately assess performance in true long-context scenarios.

W3) For efficiency analysis, direct comparisons of first-token latency and average decoding speed against full cache or methods like HeadKV would provide more straightforward evaluation metrics.

W4) In Section 3 (Sparse Attention Kernel), presenting the computation process through formal equations or algorithms would enhance clarity.

[1]Fu Y, Cai Z, Asi A, et al. Not all heads matter: A head-level kv cache compression method with integrated retrieval and reasoning[J]. ICLR 2025.

[2]Feng Y, Lv J, Cao Y, et al. Ada-kv: Optimizing kv cache eviction by adaptive budget allocation for efficient llm inference[J]. arXiv preprint arXiv:2407.11550, 2024.

[3]Li Y, Huang Y, Yang B, et al. Snapkv: Llm knows what you are looking for before generation[J]. Advances in Neural Information Processing Systems, 2024, 37: 22947-22970.

---

> ### Author Rebuttal · Authors · 2025-07-30
>
> We thank the reviewer for the constructive feedback.
>
> > **W1) The baseline comparison is limited to ThinK, while omitting recent highly relevant methods in KV cache compression such as HeadKV [1]. The reported LongBench results appear inferior to HeadKV's performance, particularly since HeadKV demonstrates effectiveness even when retaining less than 5% of KV cache.**
>
> We selected ThinK as our baseline because it is, to the best of our knowledge, the only prior method that performs channel-wise (per-token) pruning. This aligns with Mustafar's core contribution and is directly comparable.
>
> In contrast, HeadKV is not a competing baseline but a complementary method, operating on a token selection axis rather than compressing the KV cache in the channel-dimension. We view Mustafar as *orthogonal* to such token-level eviction methods, and in fact, Mustafar is explicitly designed to be combined with them.
>
> To demonstrate this, Section 4.2 includes joint application example with token eviction method of H2O. HeadKV can be seen as an advanced form of token eviction, performing fine-grained head-specific token retention. Accordingly, we present the results of combining Mustafar with HeadKV on Llama-3-8B-Instruct. For evaluation, we used one representative task from each LongBench category: single-document QA (NarrativeQA), multi-document QA (HotpotQA), summarization (GovReport), few-shot learning (TREC), synthetic (Pcount), and code (LCC).
>
>
> | Method                        | NrtvQA | HotpotQA | GovReport | TREC | Pcount | LCC   |
> |------------------------------|-------------|----------|------------|------|--------|-------|
> | HeadKV [KV size = 128]                 | 22.33       | 43.66    | 21.67      | 72.00 | 5.67   | 59.39 |
> | HeadKV + MustafarK0.5 V0.5       | 23.24       | 42.61    | 20.56      | 71.50 | 5.92   | 58.00 |
> | HeadKV + MustafarK0.7 V0.7       | 23.45       | 39.07    | 20.18      | 64.50 | 5.94   | 54.26 |
>
>
> The table above shows minimal accuracy loss when applying Mustafar's 50% sparsity on top of HeadKV's aggressive 128-token budget (1.5% of dense size), achieving additional compression to 0.975% of dense KV size. With 70% sparsity, Mustafar achieves further compression to 0.675%. These results reinforce our claim that Mustafar enables further compression beyond what structured or token-level techniques achieve alone, and should be seen as an enabling building block for such methods, rather than an alternative baseline.
>
> > **W2) The evaluation lacks standard long-context benchmarks like "Needle in a Haystack", which have become prevalent in KV cache compression research [1-3]. LongBench's average sequence length may not adequately assess performance in true long-context scenarios.**
>
> Acknowledging the need for additional benchmarks that span a longer context length, we report accuracy of Llama-3.1-8B-Instruct on RULER [1] with a context length of 65536. We note that RULER contains “Needle in a Haystack”.
>
> | Sparsity         | Method     | Needle-Single1 | Needle-Single2 | Needle-MultiKey1 | Needle-MultiKey2 | Needle-MultiQuery | Needle-MultiValue | QA-1  | QA-2  | Variable Tracking | Frequent Words Extraction |
> |------------------|------------|----------------|----------------|------------------|------------------|-------------------|-------------------|-------|-------|--------------------|----------------------------|
> | Dense            | —          | 1.00           | 1.00           | 0.990    | 0.979    | 0.990    | 0.979  | 0.844 | 0.594 | 0.973  | 0.851                      |
> | Key 50%          | ThinK      | 1.00           | 1.00           | 0.990    | 0.979  | 0.995   | 0.969  | 0.833 | 0.594 | 0.919  | 0.854     |
> |   | Mustafar   | 1.00           | 1.00           | 0.990 | 0.979  | 0.995  | 0.996   | 0.833 | 0.573 | 0.971              | 0.813        |
> | Key 70%          | ThinK      | 0.448          | 0.490          | 0.229            | 0.188            | 0.646             | 0.487             | 0.615 | 0.510 | 0.208              | 0.427                      |
> |                  | Mustafar   | 1.00  | 1.00  | 0.990  | 0.969  | 0.992   | 0.903   | 0.833 | 0.594 | 0.966              | 0.823    |
> | Value 50%        | ThinK      | 1.00           | 1.00           | 0.990  | 0.969  | 0.914   | 0.958  | 0.823 | 0.573 | 0.910   | 0.792    |
> |     | Mustafar   | 1.00   | 1.00  | 0.979    | 0.995   | 0.995   | 0.971    | 0.833 | 0.604 | 0.983   | 0.830   |
> | Value 70%  | ThinK | 0.948   | 0.927  | 0.948  | 0.510   | 0.698 | 0.688  | 0.646 | 0.500 | 0.558 | 0.677   |
> |    | Mustafar   | 1.00  | 1.00  | 1.00 | 0.979    | 0.992 | 0.969  | 0.833 | 0.594 | 0.985  | 0.826  |
> | Key&Value 50%    | ThinK      | 0.958 | 1.00 | 0.948  | 0.854 | 0.828 | 0.956  | 0.740 | 0.531 | 0.742  | 0.823   |
> |  | Mustafar   | 1.00  | 1.00 | 0.990 | 0.979 | 0.997   | 0.997  | 0.833 | 0.573 | 0.862  | 0.809 |
> | Key&Value 70%    | ThinK      | 0.000 | 0.073  | 0.000 | 0.000  | 0.000 | 0.000  | 0.219 | 0.250 | 0.000  | 0.035  |
> |    | Mustafar   | 1.00  | 1.00  | 0.990  | 0.969 | 0.995 | 0.914 | 0.833 | 0.583 | 0.869 | 0.799 |
>
> Even in challenging “Needle in a Haystack” scenarios with multiple keys and queries, Mustafar maintains accuracy comparable to dense LLM. Also, it generally outperforms the structured pruning baseline ThinK, with particularly significant gains at joint key-value 70% sparsity.
>
> > **W3) For efficiency analysis, direct comparisons of first-token latency and average decoding speed against full cache or methods like HeadKV would provide more straightforward evaluation metrics.**
>
> As discussed in our response to W1, Mustafar is orthogonal to token-level eviction methods such as HeadKV, and is designed to be complementary rather than directly competitive. Therefore, in our efficiency analysis, we focus on comparisons against the dense KV cache, which provides a more appropriate baseline to measure the impact of Mustafar’s unstructured compression. We compare against dense decoding using FlashAttention on Llama-2-7B and Llama-3-8B-Instruct with batch size 4. For Llama-2, we used input sequence length of 2048. For Llama-3, we used input sequence length of 4096. We use output sequence length of 512, 1024, and 2048.
>
> We derived the average decoding speed by measuring the end-to-end duration, and dividing it to the number of tokens generated to penalize Mustafar with the overhead of KV cache pruning and compression in both prefill and decode stages.
>
> |           Model            |       KV Format       |   TTFT   | Decode Speed (decode 512) | Decode Speed (decode 1024) | Decode Speed (decode 2048) |
> |---------------------------|------------------------|----------|-----------------------------|------------------------------|------------------------------|
> | Llama2               | Dense                  | 1.396 sec | 88.685 tokens / sec         | 88.512 tokens / sec          | 79.185 tokens / sec          |
> |                           | Mustafar K0.5 V0.5        | 2.532 sec | 89.452 tokens / sec         | 89.514 tokens / sec          | 85.687 tokens / sec          |
> |                           | Mustafar K0.7 V0.7        | 2.249 sec | 96.386 tokens / sec         | 97.436 tokens / sec          | 95.120 tokens / sec          |
> | Llama3               | Dense                  | 2.769 sec | 61.993 tokens / sec         | 61.220 tokens / sec           | 59.242 tokens / sec          |
> |                           | Mustafar K0.5 V0.5        | 3.269 sec | 78.434 tokens / sec         | 83.768 tokens / sec          | 83.303 tokens / sec          |
> |                           | Mustafar K0.7 V0.7%        | 3.151 sec | 84.065 tokens / sec         | 88.293 tokens / sec          | 89.699 tokens / sec          |
>
> While time-to-first-token is delayed due to the overhead of pruning and compressing the KV cache during the prefill stage, the delay is offset by the accelerated attention computation during decoding, resulting in higher overall token generation throughput. Notably, Llama-3 exhibits a larger performance gain compared to Llama-2, as its GQA architecture reduces the overhead of KV cache pruning and compression.
>
> > **W4) In Section 3 (Sparse Attention Kernel), presenting the computation process through formal equations or algorithms would enhance clarity.**
>
> Thank you for this suggestion. As we specify the pruning algorithms in Section 2, we now include a concise sequence of formal equations that detail how our kernel operates on the compressed sparse format. For brevity, we omit the per-head decomposition and normalization step, which follow the standard transformer formulation.
>
> *Let:*
>
>  Query vector at decoding step $ t $: $ \mathbf{Q}_t \in \mathbb{R}^d $,
>
> Key vectors within the local window: $ \mathbf{K}_L \in \mathbb{R}^{d \times N_d}, \text{where } N_d $ is the size of local window in tokens.
>
> Compressed Key vectors outside the local window: $ \mathbf{K}_C \in \mathbb{R}^{d \times N_s}, \text{where } N_s $ is the number of compressed tokens.
>
> Corresponding Value caches:
> $ \mathbf{V}_L \in \mathbb{R}^{d \times N_d} , \mathbf{V}_C \in \mathbb{R}^{d \times N_s} $
>
> *Attention score computation:*
>
> Dense local window attention score: $ \mathbf{S}_L = \mathbf{Q}_t \cdot \mathbf{K}_L , \in \mathbb{R}^{1 \times N_d} $
>
> Sparse attention score over pruned KV cache: $ \mathbf{S}_C = \mathbf{Q}_t \cdot \mathbf{K}_C , \in \mathbb{R}^{1 \times N_s} $
>
> Full attention weight $ \mathbf{S}_t = \text{softmax}(\text{concat}(\mathbf{S}_C, \mathbf{S}_L)) , \in \mathbb{R}^{1 \times (N_s + N_d)} $
>
> *Final Output computation:*
>
> $ [\mathbf{S}_C, \mathbf{S}_L] = \text{split}(\mathbf{S}_t ; N_s, N_d) $
>
> $ \mathbf{O}_t = \mathbf{V}_C \cdot \mathbf{S}_C^\top  + \mathbf{V}_L \cdot \mathbf{S}_L^\top, \in \mathbb{R}^{d} $
>
> References:
>
> [1] Cheng-Ping Hsieh, Simeng Sun, Samuel Kriman, Shantanu Acharya, Dima Rekesh, Fei Jia, Yang Zhang, Boris Ginsburg. *RULER: What's the Real Context Size of Your Long-Context Language Models?*

---

> > ### Comment · Reviewer_PxDY · 2025-08-05
> >
> > Thank you for your reply, which addressed most of my concerns. I will increase the score.

---

### Official Review · Reviewer_uxWr · 2025-07-04

**Clarity:** 3
**Significance:** 3
**Originality:** 3
**Rating:** 5
**Confidence:** 4

**Summary:**

This paper proposes an unstructured-sparsity framework for the KV cache pruning during LLM inference, which is an area that has not been well studied so far. Two key empirical observations are provided: (1) Key cache activations contain prominent channel-wise outliers; (2) Value cache, despite its near-uniform distribution, still tolerates simple per-token magnitude pruning without compromising accuracy. Based on these findings, the proposed KV cache is stored in a bitmap-based sparse format and processed on-the-fly with a custom CUDA SpMV kernel, allowing attention to operate directly on the compressed data. Experiments on LongBench with several open-source LLMs show that unstructured pruning achieves up to 70 % sparsity while matching—or surpassing—the accuracy of a structured-pruning baseline (ThinK), all without any fine-tuning.

**Questions:**

Please see weaknesses.

**Ethical Concerns:**

["NO or VERY MINOR ethics concerns only"]

**Final Justification:**

I support this paper to be accepted. This paper is the first to systematically compare unstructured and structured KV cache pruning, showing the superior accuracy–compression trade-off of unstructured methods. I think the contribution of this paper is timely and practically valuable.

**Quality:**

2

**Strengths And Weaknesses:**

Strengths:
1. The paper is the first to systematically compare unstructured KV cache pruning against structured methods, highlighting its practical superiority in accuracy–compression trade-offs.
2. The proposed method achieves up to 70% unstructured sparsity without accuracy loss, while simultaneously reducing memory footprint and improving throughput (up to 2.2×) via a custom SpMV kernel.

Weaknesses:
1. The contribution is timely and practically valuable, but the core novelty remains limited, relying on known pruning heuristics without introducing new algorithmic or theoretical insights.
2. The scalability of the method is not sufficiently validated, as experiments are limited to small-to-mid scale models (7B/8B), with no evidence that the approach holds for larger models like 13B or 70B.
3. The writing can be improved further. Some notations in the equations are undefined, and there are several typos throughout the paper—for example, a typo in Line 186. Additionally, the reported average performance of Mistral-7B-Instruct-v0.2 for K0.5 V0.5 (= 54.15) in Table 4 appears to be incorrect.

---

> ### Author Rebuttal · Authors · 2025-07-30
>
> We thank the reviewer for the constructive feedback.
>
> > **W1. The contribution is timely and practically valuable, but the core novelty remains limited, relying on known pruning heuristics without introducing new algorithmic or theoretical insights.**
>
> We appreciate the reviewer’s recognition that the contribution is timely and practically valuable. While it is true that our pruning strategy leverages well-established magnitude-based heuristics, we believe that our key contribution lies in the novel observation, empirical analysis, and system-level integration of unstructured token-wise sparsity for KV cache compression, which is an area that prior work has left largely unexplored due to its execution inefficiency.
>
> Specifically, our pruning strategy is rooted in critical empirical and theoretical findings:
> (i) Key cache exhibits channel-wise outliers, as originally suggested by KIVI. We extend this insight by demonstrating that per-token unstructured pruning is more effective at capturing these outliers than structured pruning.
> (ii) Value cache, on the other hand, shows no strong structural cues, but we reveal that token-wise magnitude pruning is inherently output-aware due to the attention formulation (Attention Score × Value), where every element contributes proportionally to the output.
>
> Furthermore, our work is the first to demonstrate that unstructured pruning of both Key and Value caches to 70% sparsity is feasible. This insight alone substantially pushes the boundary of what is considered achievable in KV cache sparsity, which had been limited to structured sparsity. Unstructured sparsity is then made practical and efficient with a custom CUDA attention kernel that efficiently operates on a bitmap-based compression format tailored for arbitrary sparsity patterns. We show that the speedup from sparse computation compensates for pruning overhead, enabling faster inference, KV cache compression, and longer context lengths without degrading performance.
>
> Finally, we validate that our pruning method is complementary to orthogonal techniques such as quantization and token eviction (Section 4.2), showing that Mustafar can be jointly applied in practical deployment scenarios. We believe this holistic treatment, from sparsity observations to kernel-level execution, constitutes a substantial novelty, both in terms of theoretical insight and practical system design.
>
> > **W2. The scalability of the method is not sufficiently validated, as experiments are limited to small-to-mid scale models (7B/8B), with no evidence that the approach holds for larger models like 13B or 70B.**
>
> We provided the evaluation for LLaMA-2-13B-Chat in Appendix A.2. While we were not able to experiment with a 70B model due to resource constraints, as ThinK demonstrated the accuracy preservation of LLaMa-3-70B-Instruct with 40% and 50% sparsity structured pruning of Key cache (ThinK paper, Table 2), we expect our method to generalize similarly.
>
> > **W3. The writing can be improved further. Some notations in the equations are undefined, and there are several typos throughout the paper—for example, a typo in Line 186. Additionally, the reported average performance of Mistral-7B-Instruct-v0.2 for K0.5 V0.5 (= 54.15) in Table 4 appears to be incorrect.**
>
> Thank you for pointing this out. We will correct the typos in the revised version, as well as clarify the equations. Also, the correct average entry for Mistral-7B-Instruct-v0.2 for K0.5 V0.5 on Table 4 is 42.30.

---

> > ### Comment · Reviewer_uxWr · 2025-08-05
> > **Response to author rebuttal**
> >
> > Thank you for your response. I will raise my score, as I believe the results of the paper are worth publishing.

---

### Official Review · Reviewer_CYDN · 2025-07-05

**Clarity:** 3
**Significance:** 4
**Originality:** 3
**Rating:** 5
**Confidence:** 4

**Summary:**

This paper introduces MUSTAFAR, a method for KV cache compression in large language models using unstructured sparsity. The key contributions are (1) demonstrating that unstructured per-token magnitude-based pruning acheive better compression ratios compared with structured pruning. (2) developing a bitmap-based sparse KV sparsity. (3) custom sparse attention kernel.
This paper makes contribution based on ThinK with unstructured sparsity and practical system implementation to improve their drawback on V-cache sparsity.

**Questions:**

See above.

**Ethical Concerns:**

["NO or VERY MINOR ethics concerns only"]

**Final Justification:**

The author addresses most of my concerns so I decide to keep the original score leaning towards acceptance.

**Limitations:**

Yes

**Quality:**

3

**Strengths And Weaknesses:**

Strengths and Contributions
- The biggest contribution of the paper imo is not the algorithm design but demonstrating it is possible to be faster than dense (which sparse method similar to quantization before usually can save memory size but way slower). While the quantization community has made a great process on the usability and practicality of quantization methods, we haven't seen sparsity made similar progress. Therefore, I think this is an important step towards unleashing the potential of unstructured sparsity in general.
- Also it is important to show how large the potential benefits of unstructured sparse. In fact, I think the authors did a good job proving it's complementary to other structured sparse and quantization methods.

Limitations and Weaknesses
- Evaluation Benchmark: The evaluation benchmarks is a bit limited (LongBench V1 only for long context evaluation). I think to convince the wider audience the power of unstructured sparse, I think we can evaluate on newer and thorough evals. We recommend using  standard ones like Ruler [1] for million-level benchmark and newer and harder ones like GSM-Infinite [3] for long-context reasoning benchmark.
- Discussion on dynamic sparse attention: Other than static sparse and quantization methods, there's a line of work in dynamic sparse attention such as [3]. Since it requires storage format change, how does it impact dynamic sparse methods which stores the whole kv cache on GPU or CPU? Can it still be combined?
- The paper is built based on the observation of KIVI, and make extensive development on large language model like Llama-2-7B, Llama-3-8B-Instruct, Mistral-7B. If the observation can be applied on recent reasoning models like deepseek[7] or Qwen[8], that would greatly enhance the impact of the paper.

[1] Cheng-Ping Hsieh, Simeng Sun, Samuel Kriman, Shantanu Acharya, Dima Rekesh, Fei Jia, Yang Zhang, Boris Ginsburg. RULER: What's the Real Context Size of Your Long-Context Language Models?

[2] Yang Zhou, Hongyi Liu, Zhuoming Chen, Yuandong Tian, Beidi Chen. GSM-Infinite: How Do Your LLMs Behave over Infinitely Increasing Context Length and Reasoning Complexity?

[3] Zhuoming Chen, Ranajoy Sadhukhan, Zihao Ye, Yang Zhou, Jianyu Zhang, Niklas Nolte, Yuandong Tian, Matthijs Douze, Leon Bottou, Zhihao Jia, Beidi Chen. MagicPIG: LSH Sampling for Efficient LLM Generation.

[4] Liu A, Feng B, Xue B, et al. Deepseek-v3 technical report[J]. arXiv preprint arXiv:2412.19437, 2024.

[5] Yang A, Li A, Yang B, et al. Qwen3 technical report[J]. arXiv preprint arXiv:2505.09388, 2025.

Additional Feedback:
Typos
- Line 206 sparisty->sparsity
- Figure 5: Overview of Mustafar attention fernel -> Kernel
- Line 283 FlashDecoding[15] introduces double-buffering do -> to

---

> ### Author Rebuttal · Authors · 2025-07-31
>
> We thank the reviewer for the constructive feedback.
>
> > **W1. Evaluation Benchmark: The evaluation benchmarks is a bit limited (LongBench V1 only for long context evaluation). I think to convince the wider audience the power of unstructured sparse, I think we can evaluate on newer and thorough evals. We recommend using standard ones like Ruler [1] for million-level benchmark and newer and harder ones like GSM-Infinite [3] for long-context reasoning benchmark.**
>
> We had selected LongBench as it covers various use scenarios from question-answering to summarization. Also, version 1 was the primary benchmark of the previous works KIVI and ThinK. Following the recommendation of the reviewer, we evaluated Mustafar on Llama-3.1-8B-Instruct, with Ruler for context length of 65536.
>
> | Sparsity         | Method     | Needle-Single1 | Needle-Single2 | Needle-MultiKey1 | Needle-MultiKey2 | Needle-MultiQuery | Needle-MultiValue | QA-1  | QA-2  | Variable Tracking | Frequent Words Extraction |
> |------------------|------------|----------------|----------------|------------------|------------------|-------------------|-------------------|-------|-------|--------------------|----------------------------|
> | Dense            | —          | 1.00           | 1.00           | 0.990            | 0.979            | 0.990             | 0.979             | 0.844 | 0.594 | 0.973              | 0.851                      |
> | Key 50%  | ThinK  | 1.00  | 1.00  | 0.990 | 0.979   | 0.995  | 0.969   | 0.833 | 0.594 | 0.919 | 0.854  |
> |  | Mustafar   | 1.00  | 1.00  | 0.990  | 0.979 | 0.995  | 0.996  | 0.833 | 0.573 | 0.971  | 0.813  |
> | Key 70%  | ThinK | 0.448  | 0.490  | 0.229 | 0.188 | 0.646  | 0.487  | 0.615 | 0.510 | 0.208  | 0.427  |
> |  | Mustafar   | 1.00 | 1.00 | 0.990  | 0.969 | 0.992 | 0.903  | 0.833 | 0.594 | 0.966 | 0.823 |
> | Value 50%  | ThinK | 1.00 | 1.00 | 0.990  | 0.969  | 0.914   | 0.958  | 0.823 | 0.573 | 0.910   | 0.792    |
> |     | Mustafar   | 1.00   | 1.00  | 0.979    | 0.995   | 0.995   | 0.971    | 0.833 | 0.604 | 0.983   | 0.830   |
> | Value 70%  | ThinK | 0.948   | 0.927  | 0.948  | 0.510   | 0.698 | 0.688  | 0.646 | 0.500 | 0.558 | 0.677   |
> |    | Mustafar   | 1.00  | 1.00  | 1.00 | 0.979    | 0.992 | 0.969  | 0.833 | 0.594 | 0.985  | 0.826  |
> | Key&Value 50%    | ThinK      | 0.958 | 1.00 | 0.948  | 0.854 | 0.828 | 0.956  | 0.740 | 0.531 | 0.742  | 0.823   |
> |  | Mustafar   | 1.00  | 1.00 | 0.990 | 0.979 | 0.997   | 0.997  | 0.833 | 0.573 | 0.862  | 0.809 |
> | Key&Value 70%    | ThinK      | 0.000 | 0.073  | 0.000 | 0.000  | 0.000 | 0.000  | 0.219 | 0.250 | 0.000  | 0.035  |
> |    | Mustafar   | 1.00  | 1.00  | 0.990  | 0.969 | 0.995 | 0.914 | 0.833 | 0.583 | 0.869 | 0.799 |
>
> Even in challenging “Needle in a Haystack” scenarios with multiple keys and queries, Mustafar maintains accuracy comparable to dense LLMs. It also generally outperforms the structured pruning baseline ThinK, with particularly notable gains at 70% joint Key-Value sparsity. While structured pruning does perform well in isolated cases, such as the Needle-Single tasks for 70% Value sparsity, it exhibits significant accuracy drops in other tasks. In contrast, Mustafar’s unstructured sparsity consistently preserves accuracy across all tasks. This contrast highlights the versatility of unstructured sparsity in adapting to diverse task requirements.
>
> We provide GSM-Infinite evaluation in our response for W3.
>
> > **W2. Discussion on dynamic sparse attention: Other than static sparse and quantization methods, there's a line of work in dynamic sparse attention such as [3]. Since it requires storage format change, how does it impact dynamic sparse methods which stores the whole kv cache on GPU or CPU? Can it still be combined?**
>
> We first highlight the motivation behind Mustafar: GPU memory is a heavily contended resource due to the storage demands of both model weights and the KV cache. Mustafar addresses this by compressing the KV cache in GPU memory, thereby alleviating pressure on memory capacity. It further bridges the gap between the massively parallel compute capability of GPU microarchitecture and the memory-bound nature of attention computation during decoding. This is achieved by transferring the KV cache in its compressed form from global memory and directly computing on high-throughput Tensor Cores.
>
> In contrast, MagicPIG leverages the larger and more flexible CPU memory to store the majority of KV cache, minus sink and local tokens. To compensate for the CPU’s limited compute throughput, it reduces the overall attention cost via a sampling-based attention scheme where selective tokens' KV cache is used for computation.
>
> Regarding support for a sampling-based attention scheme on GPU, Mustafar’s sparse format and attention kernel are readily extensible to support such sampling-based attention schemes. This would requires modest modifications to the sparse format and indexing logic within the CUDA attention kernel, enabling 64-element tiles to be created per-token (as opposed to the current across-token tiling shown in Figure 8). Since the number of tiles per token is fixed for a given LLM architecture, the attention kernel can treat this as a static stride and use it to efficiently locate and compute only the relevant tiles under the same pipeline as described in Figure 7. Similarly, this enables managing KV cache in token-granularity to offload to CPU memory.
>
> For CPU execution, however, the pipeline in Figure 7 is not directly applicable, as Mustafar depends on the GPU’s addressable shared memory (L1 cache) to stage decompressed tiles. In the absence of such an addressable cache on CPUs, decompression and computation must be fused and performed while operands are in CPU registers. Nevertheless, compression of KV cache will mitigate the bottleneck on CPU memory bandwidth, which is ‘usually 10 − 20% of GPU VRAM bandwidth’ (MagicPIG, Section 4.4).
>
> Finally, for hybrid CPU-GPU setups (e.g., FlexGen[1] and InfiniGen[2]) where the KV cache is stored on the CPU and transferred to the GPU for attention computation, Mustafar still offers advantages: by reducing the memory footprint of the KV cache, it proportionally reduces the data transfer time between CPU and GPU.
>
> > **W3. The paper is built based on the observation of KIVI, and make extensive development on large language model like Llama-2-7B, Llama-3-8B-Instruct, Mistral-7B. If the observation can be applied on recent reasoning models like deepseek[7] or Qwen[8], that would greatly enhance the impact of the paper.`**
>
> We visualized the KV cache of Deepseek-R1-0528-Qwen3-8B and Deepseek-R1-Distill-Llama-8B (selected in place of the 685B parameter Deepseek-v3 due to resource constraints). Across both models, we observed a similar pattern: the Key cache exhibits an outlier-heavy distribution, whereas the Value cache displays a more uniform distribution.
>
> To evaluate the effectiveness of Mustafar in reasoning models on reasoning tasks, we conducted experiments on Deepseek-R1-0528-Qwen3-8B using a down-sized GSM-Infinite benchmark. Due to time constraints, we limited our evaluation to the zero-noise setting, with an operation range of 1–50 for Symbolic task and 2–30 for Medium and Hard tasks, yielding 24 to 65 samples per task. We report Area Under the Curve (AUC) as the evaluation metric.
>
> | Sparsity        | Method   | Symbolic | Medium | Hard |
> |-----------------|----------|----------|--------|------|
> | Dense           | -        | 1520     | 625    | 275  |
> | Key&Value 50%   | Mustafar | 1520     | 750    | 375  |
> | Key&Value 70%   | Mustafar | 240      | 187.5  | 50   |
> | Value 70%       | ThinK    | 0        | 62.5   | 100  |
>
> While 70% structured pruning of Value cache performs relatively well on the Hard task, it completely fails on Symbolic and struggles on Medium. This pattern mirrors our KV cache visualization, where the Value cache exhibits a relatively uniform distribution, leading to inconsistent accuracy across task types under structured sparsity .
>
> In contrast, at 50% unstructured sparsity applied to both Key and Value caches, Mustafar retains performance across all GSM-Infinite tasks. While accuracy degrades when sparsity is increased to 70%, Mustafar still maintains some level of accuracy across all task types, unlike structured pruning. These results demonstrate that Mustafar’s benefits extend to reasoning models. Although our current evaluation focuses on unstructured pruning with uniform sparsity, we believe that incorporating component-wise importance-based sparsity, as illustrated in our toy example response to QfM2 (Q1), offers a promising direction to increase accuracy under high sparsity. Nonetheless, the presented results affirm Mustafar's effectiveness in reasoning scenarios.
>
>
> > **Additional Feedback: Typos**
>
> Thank you for pointing this out. We will correct the typos in the revised version.
>
> References:
>
> [1] Ying Sheng, Lianmin Zheng, Binhang Yuan, Zhuohan Li, Max Ryabinin, Daniel Y. Fu, Zhiqiang Xie, Beidi Chen, Clark Barrett, Joseph E. Gonzalez, Percy Liang, Christopher Ré, Ion Stoica, Ce Zhang. *FlexGen: High-Throughput Generative Inference of Large Language Models with a Single GPU.*
>
> [2] Wonbeom Lee, Jungi Lee, Junghwan Seo, Jaewoong Sim. *InfiniGen: Efficient Generative Inference of Large Language Models with Dynamic KV Cache Management.*

---

> > ### Comment · Reviewer_CYDN · 2025-08-05
> > **Concerns resolved**
> >
> > Thanks authors for the detailed response and added GSMinf results! I'll keep my support.

---

### Comment · Area_Chair_6M1K · 2025-08-02

Dear Reviewers

The authors have responded to your reviews. In the next few days, please read their responses and engage in a productive discussion that will be critical to the review process.

I truly appreciate your timely thoughts and comments!

AC

---

### Note · Authors · 2025-08-11

Dear Reviewers and Area Chairs,

We appreciate the insightful and constructive comments from all reviewers. We are glad that the significance of Mustafar, as the first work to promote unstructured sparsity for KV cache compression and computational efficiency in LLMs, was recognized. According to all reviewers' final confirmation, we believe we have addressed all feedback to every reviewer's satisfaction.

- **Reviewer CYDN**: We conducted additional evaluations on longer-context and reasoning tasks as requested, including RULER, GSM-Infinite, and reasoning models Deepseek-Qwen and Deepseek-Llama. We also discussed the benefits of Mustafar in supporting dynamic sparse attention.
- **Reviewer uxWr**: We clarified the novelty of Mustafar, emphasizing its core contributions in empirical observation, theoretical findings, and system-level integration of unstructured sparsity in the KV cache. We also demonstrated its scalability to larger models.
- **Reviewer PxDY**: We compared Mustafar with HeadKV, provided the rationale for their orthogonality, and showed that Mustafar can further compress the aggressively evicted KV cache of HeadKV while preserving accuracy. We added additional long-context evaluations with RULER, reported efficiency metrics in TTFT and decode speed, and included formal equations for the Mustafar Sparse Attention Kernel.
- **Reviewer QfM2**: We explained the advantages of the Mustafar bitmap format in low-sparsity regimes, clarified why token-wise eviction methods like HeadKV are better viewed as complementary rather than competing baselines, and validated the benefit of considering head-wise importance through toy experiments, enabled by Mustafar’s ability to represent arbitrary sparsity levels.

Overall, we found the reviewer interactions highly valuable and will incorporate these outcomes into the final version of Mustafar.

---

### Decision · Program_Chairs · 2025-09-17

**Decision:**

Accept (poster)

**Comment:**

After a fruitful discussion that addressed many of the reviewers' concerns, the consensus is that this is a nice paper with results that would be of interest to the community. Thus, I recommend acceptance.